# APIC: Orthogonalized Neuro-Symbolic Modeling for Nonlinear Dissipative Dynamics

**Yanhui Zhu** [1]   **Xiangfu Meng** [1] [*]   **Chen Zhao** [1]   **Yinhao Li** [1]

## Abstract

Current data-driven scientific modeling struggles with a functional dichotomy: neural operators exhibit spectral bias in high-frequency regimes, while physics-constrained paradigms suffer from optimization pathologies. To bridge this gap, we propose Adaptive Physics-Informed Computing (APIC), a neuro-symbolic meta-architecture designed with structural reconfigurability to encode diverse domain priors. Crucially, APIC integrates a gradient isolation strategy that reduces interference between the optimization paths of parameter identification and residual correction, effectively mitigating gradient conflicts. By instantiating this framework for nonlinear dissipative systems, we derive the Generalized Kuramoto-Sivashinsky-Cahn-Hilliard (G-KSCH) kernel, providing a unified representation for sparse dynamic identification. Extensive experiments demonstrate that APIC establishes new benchmarks in 3D compressible supersonic shock wave prediction, surpassing diverse architectures (e.g., CNNs and Transformers) by substantial margins in predictive accuracy. Notably, APIC achieves Pareto-optimal performance, delivering superior precision with reduced computational overhead compared to SOTA models, while exhibiting strong cross-task adaptability across meteorological and urban traffic datasets.

## 1. Introduction

While deep learning is increasingly employed to address scientific challenges that are computationally intractable for traditional numerical methods, current architectures are predominantly adopted from Computer Vision or Natural Language Processing. In this data-driven paradigm, continuous physical fields are discretized into static representations, such as pixels or tokens, often neglecting the underlying governing equations and conservation laws. This results in a fundamental structural mismatch: while existing models function as effective interpolators within the training distribution, their generalizability significantly deteriorates in out-of-distribution scenarios where strict adherence to causality and physical constraints is essential.

Current solutions generally fall into a tripartite dilemma, with no single approach simultaneously satisfying accuracy, efficiency, and physical consistency:

- **Physics-agnostic Data-Driven Models:** Represented by Transformers (Vaswani et al., 2017) and CNNs, these methods face an intrinsic trade-off between generalization and efficiency. CNNs are restricted by local receptive fields, failing to capture global pressure waves. Transformers, despite global modeling capabilities, lack inductive bias; learning spatial continuity demands substantial parameter scaling or excessive memory consumption.
- **Neural Operators:** Approaches like FNO (Li et al., 2021) and Transolver (Wu et al., 2024), while theoretically defining mappings between function spaces, suffer from extreme data inefficiency. Learning complex operators such as high-Reynolds number turbulence from scratch requires massive, expensive simulation data, effectively shifting the computational burden from the inference phase to the data generation phase.
- **Physics-Priors Approaches:** Approaches to incorporating physical priors face intrinsic trade-offs: **structurally rigid** designs often falter under real-world noise (Rao et al., 2023), while **regularization-based** methods suffer from optimization pathologies and spectral bias due to gradient conflicts (Raissi et al., 2019; Wang et al., 2021). Similarly, **implicit** architectures rely on opaque operators that lack the explicit semantics necessary to capture strong non-linearities like shock waves (Le Guen & Thome, 2020).

To bridge this gap, we propose Adaptive Physics-Informed Computing (APIC). Unlike methods that treat physics

---

[*]Corresponding author. [1]School of Electronic and Information Engineering, Liaoning Technical University, Huludao, Liaoning, China. Correspondence to: Xiangfu Meng <mengxiangfu@lntu.edu.cn>.

*Proceedings of the 43rd International Conference on Machine Learning*, Seoul, South Korea. PMLR 306, 2026. Copyright 2026 by the author(s).

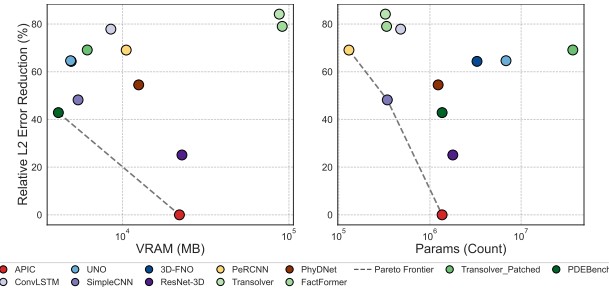

*Figure 1.* Pareto Optimality Analysis on High-Dimensional CFD Dynamics. The scatter plots illustrate the Relative L2 Error Reduction against VRAM usage and Parameter counts. APIC establishes a new state-of-the-art on the Pareto frontier, significantly outperforming existing baselines in both accuracy and resource efficiency.

merely as an auxiliary regularization term, APIC introduces a Gradient Isolation Mechanism. We deconstruct the learning process into two streams with reduced gradient interference.

In the Physics Stream, to endow the model with cross-task inductive bias, we construct an Adaptive Operator Slot. By introducing learnable adaptive coefficients, this slot functions as a dynamic confidence gating mechanism, autonomously arbitrating the weight between the deep learning components and explicit prior formulas. Furthermore, we Synthesize a reconfigurable Generalized Kuramoto-Sivashinsky-Cahn-Hilliard (G-KSCH) operator basis. This basis adaptively reconstructs physical terms including advection, diffusion, and phase separation. This operator serves as a versatile mathematical abstraction for diverse dissipative structures, adaptively resolving non-linear evolution problems ranging from continuous fluids to discrete traffic flows via flexible parameter configuration.

In the Data Stream, to achieve an organic integration of physics and data, we design a triple interaction mechanism of feature alignment, channel concatenation, and gradient isolation. First, to address the spatiotemporal semantic misalignment, we introduce a physics adapter that performs isomorphic mapping on dual-stream features. Second, discarding linear weighting strategies that lead to information discontinuity, we employ channel concatenation with non-linear transformation (stacked fusion). Employing channel concatenation combined with non-linear convolution at both the input embedding and the final decoding stages, this mechanism empowers the network to capture non-linear dependencies between physics and data. Finally, the data backbone is designed as an independent slot, such as 3D Attention-ResNet, which, combined with a Gradient Isolation strategy, achieves gradient-isolated optimization of physical law learning and residual correction.

Overall, our contributions are summarized as follows:

- **Gradient-Isolated Neuro-Symbolic Architecture:** We propose the APIC meta-architecture, which reduces gradient interference between physical parameter identification and residual correction. It resolves neural semantic conflicts via the Physics Adapter, eliminates optimization pathologies through Gradient Isolation, and addresses linear fusion discontinuities via non-linear stacking fusion.
- **Unified G-KSCH Physical Basis:** We derive the reconfigurable G-KSCH meta-operator based on the Reynolds Transport Theorem. Acting as a physical basis for non-equilibrium systems, it allows for adaptive reconstruction of physical terms such as advection, diffusion, and phase separation, successfully enabling sparse dynamics identification across domains like fluid dynamics and traffic flow.
- **Establishing SOTA & Pareto Optimality:** APIC establishes new benchmarks in 3D shock wave prediction tasks, demonstrating superior robustness in low-data regimes by effectively constraining the hypothesis space via G-KSCH priors with only 10% or 5% data. It stands strictly on the Pareto frontier regarding memory footprint and parameter count.

## 2. Related Work

### 2.1. Physics-Agnostic Data-Driven Architectures

In the realm of AI for Science, Physics-agnostic data-driven models predominantly rely on CNN or Transformer architectures. Convolutional Neural Networks, such as 3D-ResNet (Hara et al., 2018) and SimpleCNN (LeCun et al., 1998), approximate local derivatives via convolution kernels, effectively functioning as local differential solvers. However, restricted by limited receptive fields, CNNs struggle to capture global physical processes like pressure wave propagation and tend to prioritize low-frequency components, resulting in the loss of high-frequency turbulence details—a phenomenon known as spectral bias (Rahaman et al., 2019). Conversely, Transformer architectures like Transolver (Wu et al., 2024) and FactFormer (Li et al., 2023) achieve global modeling via self-attention, essentially acting as global integral solvers. But they face two intrinsic limitations: lack of inductive bias, as they are oblivious to spatial locality and require massive data to learn spatial continuity and computational explosion, where the complexity of self-attention becomes unsustainable for high-resolution 3D meshes. Fundamentally, these architectures remain opaque "black boxes" that are agnostic to physical mechanisms.

### 2.2. Neural Operator Learning

Neural Operator Learning, represented by FNO (Li et al., 2021) and Transolver (Wu et al., 2024), aims to learn mappings between function spaces. While theoretically offering resolution invariance, in practice, these methods often suf-

fer from superfluous empirical reconstruction: expending massive computational resources to empirically fit known physical equations from data rather than leveraging the equations directly. This approach is not only data-inefficient but also lacks explicit physical constraints. In contrast, APIC avoids this redundancy by directly implanting known governing equations as prior operators within the network.

## 2.3. Paradigms of Physics Priors

Attempts to incorporate physical constraints generally fall into two categories: hard and soft constraints.

Hard constraints, such as PeRCNN (Rao et al., 2023) force network weights to adhere to specific discretized difference schemes. While guaranteeing physical consistency, this approach sacrifices the adaptive flexibility of deep learning. When real-world data contains noise or unmodeled dynamics, the rigid physical operators often act as interference, causing model failure.

Soft constraints, exemplified by PINNs (Raissi et al., 2019) suffer from optimization pathologies due to gradient conflicts between physical residuals and data fitting losses (Wang et al., 2021). Neural networks tend to prioritize simple low-frequency components, causing physical constraints to degenerate into ineffective decorations during inference.

Similarly, implicit constraints like PhyDNet (Le Guen & Thome, 2020) attempts to implicitly simulate physical processes via specialized recurrent units. However, their internal operators are still learned from data and lack explicit physical semantics, making it difficult to capture strong non-linearities such as shock waves.

## 2.4. Hybrid Physics-Data Methods

A growing body of work attempts to more tightly couple data-driven and physics-based components. Defect-correction approaches (Um et al., 2020) employ a learned neural corrector to compensate for numerical discretization errors in a coarse physics solver, improving accuracy without sacrificing interpretability. Differentiable physics frameworks (Holl et al., 2020; Kochkov et al., 2021) enable end-to-end gradient flow through differentiable PDE solvers, allowing neural networks to be trained jointly with physical simulators; however, they require differentiable solvers and incur substantial memory cost for high-dimensional systems. Equation-learning methods such as SINDy (Brunton et al., 2016) and its deep variants (Champion et al., 2019) identify sparse governing equations from data via dictionary regression, offering high interpretability but struggling with partial observability and high-dimensional chaotic fields. APIC differs from all these paradigms: rather than correcting a fixed solver or learning equations post-hoc, it embeds a re-

configurable symbolic operator basis directly as a first-class network component, using gradient isolation to prevent data-driven objectives from corrupting the physically meaningful parameters.

## 3. Methodology

### 3.1. APIC Architecture Overview

As illustrated in Figure 2, APIC adopts a "Dual-Stream Synergistic Evolution and Full-Stack Fusion" paradigm. Unlike traditional serial correction strategies, APIC deconstructs the inference process into a parallel interaction between Physics Stream and Data Stream. The core inference pipeline proceeds as follows:

**Step 1: Physics Branch Evolution.** The physics branch functions as a prior deduction engine. Taking the initial state $U_t$ as input, it computes spatial derivatives (e.g., $\nabla U$, $\nabla^2 U$) via fixed difference kernels. These derivatives are combined with learnable adaptive coefficients $\lambda$ to explicitly calculate the physical prediction state $U_{t+1}^{\text{phys}}$ via the G-KSCH meta-equation.

**Step 2: Manifold Alignment via Physics Adapter.** To facilitate the robust interaction between explicit physical states and data-driven features, we introduce a *Physics Adapter*. We first enforce **Gradient Isolation** by strictly executing a `detach` operation on the physical prediction $U_{t+1}^{\text{phys}}$; this critical step severs the backward gradient flow from the data branch, ensuring that the adaptive coefficients $\lambda$ are optimized independently based solely on physical conservation constraints. Subsequently, we perform **Manifold Mapping** by projecting both the gradient-isolated physical state and the raw input $U_t$ into a unified high-dimensional latent space via shallow embedding layers, yielding semantically aligned feature representations denoted as $Z_{\text{phys}}$ and $Z_{\text{data}}$.

**Step 3: Backbone Learning with Holistic Fusion.** Employing a *Holistic Stacked Fusion* strategy, $Z_{\text{phys}}$ and $Z_{\text{data}}$ are directly concatenated along the channel dimension. This hybrid feature tensor is fed into the *Data Backbone*, which utilizes non-linear transformations to focus on capturing high-frequency residuals, such as turbulent micro-structures, that the physical operator fails to model, producing the refined feature $Z_{\text{res}}$.

**Step 4: Hybrid Decoding.** In the decoding phase, we adopt a symmetric stacking strategy. The backbone output $Z_{\text{res}}$ is fused again with the original physical flow $U_{t+1}^{\text{phys}}$ before entering the decoder to generate the final prediction $\hat{U}_{t+1}$.

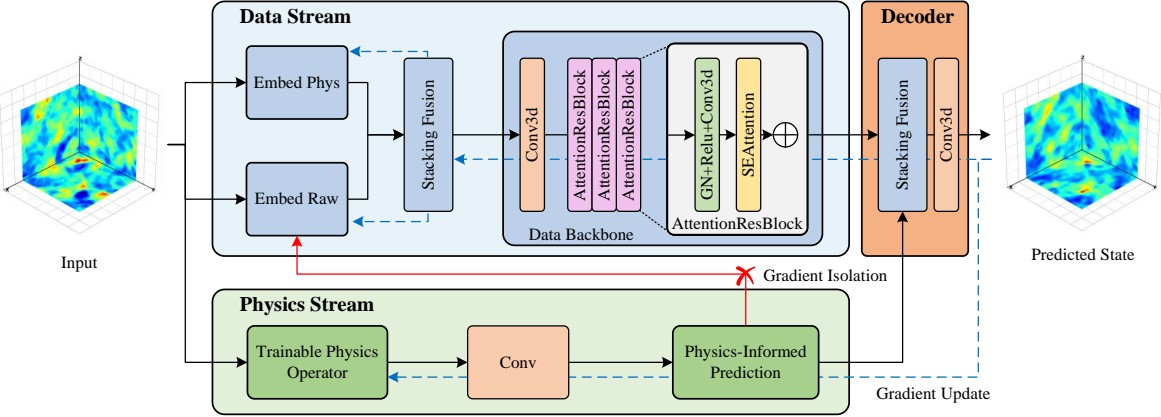

*Figure 2.* **APIC Architecture Overview.** Illustration of the Dual-Stream Synergistic Evolution and Full-Stack Fusion paradigm. The Physics Stream (top) learns explicit laws via G-KSCH, while the Data Stream (bottom) learns high-frequency residuals, interacting via the Physics Adapter.

## 3.2. Rule Operator & G-KSCH Equation

### 3.2.1. REYNOLDS TRANSPORT THEOREM (RTT)

Before constructing a universal physical operator, we examine the numerical implementation of classical dissipative systems. Whether it is the Navier-Stokes equation ($\partial_t u + u \cdot \nabla u = \nu \nabla^2 u - \nabla p$) and the Burgers equation ($\frac{\partial u}{\partial t} + u \frac{\partial u}{\partial x} = \nu \frac{\partial^2 u}{\partial x^2}$) describing fluid momentum, or the LWR model ($\partial_t \rho + \partial_x(\rho v) = 0$) for traffic flow (Lighthill & Whitham, 1955; Richards, 1956), when discretized using the Finite Difference Method (FDM), their core operators degenerate into consistent convolutional stencil forms (LeVeque, 2007) such as Sobel gradient kernels and Laplacian kernels. This numerical *Homomorphism* implies that they follow a unified underlying logic of conservation (Çengel & Cimbala, 2014).

To theoretically unify these equations, we leverage the conservation principles underlying the Reynolds Transport Theorem (RTT), specifically its implication on flux continuity. RTT states that for any control volume $\Omega(t)$ moving with the fluid, the rate of change of a physical quantity $\Psi$ equals the sum of local storage change and the net flux across the boundary $\partial\Omega$. For a fixed grid under an Eulerian perspective, its differential form is:

$$\frac{\partial u}{\partial t} + \nabla \cdot J(u) = S(u) \tag{1}$$

where $J(u)$ is the Generalized Flux and $S(u)$ is the source term. Based on this, we propose *Generalized Flux Decomposition*: asserting that the differences in dissipative systems lie solely in the physical mechanisms driving the flux $J(u)$. Consequently, we decompose $J(u)$ into a linear superposition of macroscopic advection, microscopic diffusion, phase separation flux, and high-order regularization flow.

### 3.2.2. G-KSCH OPERATOR BASIS & ADAPTIVE COEFFICIENTS

Based on the flux decomposition from RTT, we derive the Generalized Kuramoto-Sivashinsky-Cahn-Hilliard (G-KSCH) operator basis. This equation is not a simple concatenation but a unified mathematical abstraction of various dissipative structures in non-equilibrium thermodynamics:

**Nonlinear Advection:** $\mathbf{J}_{\text{adv}} = -u \cdot \nabla u$ This term is the foundation of dissipative systems, describing the macroscopic migration of matter with the flow field. To handle numerical instability at shock waves, we adopt the soft-truncated Burgers form.

**Fickian Diffusion:** $\mathbf{J}_{\text{diff}} = \nu \nabla^2 u$ Describes microscopic Brownian motion driven by concentration gradients, providing second-order smoothing and entropy increase.

**Phase Separation:** Introducing the Ginzburg-Landau chemical potential function (Ginzburg & Landau, 1950), we derive the chemical potential. The resulting term: $\mathbf{J}_{\text{sep}} = \nabla^2(u^3 - u)$ introduces a "reverse diffusion" mechanism, forcing the system to separate into sharp phase interfaces.

**Hyper-diffusion:** $\mathbf{J}_{\text{hyper}} = -\kappa \nabla^4 u$ acting as a fourth-order derivative regularization, effectively suppresses high-frequency numerical oscillations arising from the phase separation term to ensure stability.

**Reaction Source:** $\mathbf{J}_{\text{reac}} = u$ Describes processes of spontaneous generation or annihilation within the locality without boundary transport. This term is crucial for correcting mass or energy non-conservation caused by discretization errors.

It is essential to recognize that most physical models, including the Navier-Stokes and Schrödinger equations, are essentially empirical approximations of reality, where the

applicability of each term is limited to specific scales. Therefore, G-KSCH should be viewed not as a rigid formula but as a universal hypothesis space. To adapt this space to specific tasks, we introduce trainable adaptive coefficients $\lambda$ before each term. These coefficients are constrained within reasonable physical ranges via `tanh` or `sigmoid` functions—for instance ensuring diffusion coefficients remain non-negative—allowing the neural network to autonomously determine the optimal intensity of each physical mechanism via gradient descent. Combining these, we obtain the G-KSCH operator basis:

$$\frac{\partial u}{\partial t} = -\lambda_1(u \cdot \nabla u) + \lambda_2 \nabla^2 u + \lambda_3 \nabla^2(u^3 - u) - \lambda_4 \nabla^4 u + \lambda_5 u \quad (2)$$

To accommodate varying physical influences, we categorize the coefficients $\lambda_i$ into dominant ($\lambda_i = 1 + kf(\alpha_i)$) and perturbation ($\lambda_i = kf(\alpha_i)$) modes, where $f$ is the activation, $k$ is a scaling factor, and $\alpha_i$ is the learnable parameter.

### 3.2.3. CONVOLUTIONAL STENCIL IMPLEMENTATION

To efficiently solve PDEs within a deep learning framework, we implement fixed *Stencil Kernels* based on the FDM. Unlike standard learnable convolutions, the weights of these kernels are mathematically determined: First-order gradients are computed using three Sobel-like kernels with fixed weights; second-order Laplacians employ a standard 7-point difference template where the center weight is $-6$ and immediate neighbors are $1$. Furthermore, fourth-order hyper-diffusion is implemented by cascading two Laplacian convolutions, $\nabla^4 u = \nabla^2(\nabla^2 u)$, a recursive computation strategy that significantly reduces memory overhead.

### 3.3. Gradient Isolation and Physics Adapter

While the Data Stream encodes the current state $U_t$, the Physics Stream derives a **coarse** future estimate $U_{t+1}^{\text{phys}}$. Direct interaction between these representations induces a temporal semantic mismatch, as the network struggles to align the static observational features with the dynamic evolutionary priors.

To this end, APIC implements a gradient-isolated neuro-symbolic fusion through two core designs: a Gradient Isolation Strategy and a Dual-Stream Manifold Alignment. First, gradient isolation is achieved by applying a stop-gradient operation to the physics branch's prediction, creating a "shadow variable" $\hat{u}_{phys}^{\perp}$. This blocks gradient flow from the data backbone to the physical parameters $\lambda$, thereby preventing optimization conflict from data-driven noise. Second, dual-stream manifold alignment projects both the current state $u_t$ and the gradient-isolated physics estimate $\hat{u}_{phys}^{\perp}$ onto a shared semantic manifold via learnable affine projections $\Phi_{data}$ and $\Phi_{phys}$. This symmetric alignment resolves spatiotemporal mismatches between the two streams, enabling effective fusion without gradient interference.

This architecture enforces a unidirectional information flow where the data backbone treats the physical prediction as a detached reference, preventing data-driven noise from corrupting physical parameters $\lambda$. More precisely, the `detach` placement *inside* the operator argument (i.e., $\mathcal{P}(\text{detach}(\mathcal{N}_\theta(X)))$ rather than $\text{detach}(\mathcal{P}(\mathcal{N}_\theta(X)))$) removes the dominant indirect Jacobian path from the data branch back to the physics coefficients, leaving only shallow fusion interaction; a full first-order Jacobian and second-order Hessian derivation showing how this stabilises the optimisation landscape is given in Appendix B. Consequently, physical identification (law discovery) and residual correction (data fitting) follow optimization paths with reduced gradient interference, effectively mitigating gradient conflicts common in hybrid modeling.

### 3.4. Deep Stacking Interaction

Unlike traditional linear weighting or gating fusion, APIC adopts a channel stacking interaction strategy:

$$Z_{fused} = \mathcal{H}(\text{Concat}([Z_{data}, Z_{phys}])) \quad (3)$$

Existing physics fusion methods often employ shallow linear gating mechanisms:

$$u = G \cdot u_{phys} + (1 - G) \cdot u_{data}, \quad G \in [0, 1] \quad (4)$$

This linear subspace constraint creates an expressivity bottleneck, particularly when reconciling *Heterogeneous Modalities* across disparate semantic spaces.

In contrast, APIC employs Stacking Interaction, passing concatenated features $Z_{stack} = [Z_{data} \oplus Z_{phys}]$ through a non-linear transform $\mathcal{H}$. Leveraged by the *Universal Approximation Theorem* (Hornik, 1991), this empowers the model to approximate arbitrary complex fusion dynamics beyond the limits of linear superposition. Compared to linear gating, stacking fusion significantly expands the hypothesis space, enabling the network to learn complex *Conditional Logic*. This non-linear logic is inexpressible by simple linear weighting, effectively resolving the issue of information discontinuity in linear fusion.

### 3.5. Meta-Architecture Slotting

APIC adopts a modular decoupled design consisting of two independent slots to ensure cross-task adaptability. For the **Backbone Slot**, we employ a 3D SE-ResNet (Hara et al., 2018; He et al., 2016; Hu et al., 2018) as the default architecture to reconcile spatial inductive biases with long-range dependencies, recalibrating feature maps $\mathbf{X}$ via an embedded gating mechanism $\mathbf{s} = \sigma(\mathbf{W}_2 \delta(\mathbf{W}_1 \text{avgpool}(\mathbf{X})))$. Complementarily, the **Physics Operator Slot** facilitates structural adaptation to non-traditional domains. For instance, in

urban traffic modeling (Zhang et al., 2017), we retain standard transport terms (Lighthill & Whitham, 1955; Richards, 1956) while reinterpreting the phase separation operator as a *Congestion Potential* $\nabla^2(\rho^2)$ and expanding the reaction scope to $[-C, C]$ to capture rapid source/sink dynamics. This explicit, code-based customization enables APIC to seamlessly transfer across distinct scientific manifolds with exceptional architectural generalizability.

# 4. Experiments

## 4.1. Experimental Setup

### 4.1.1. DATASETS AND BENCHMARKS

We evaluate APIC across a diverse spectrum of dynamic regimes—ranging from explicit microscopic fluids to latent macroscopic systems—to scrutinize its robustness. This multi-scale setup rigorously tests the model's capacity to capture strong non-linear discontinuities under strict conservation laws, while simultaneously assessing its reconfigurability in quasi-physical scenarios where governing priors are incomplete.

Specifically, we utilize three standard benchmarks: 1) **Compressible CFD (PDEBench)** (Takamoto et al., 2022): A 3D transonic turbulence dataset (3D_CFD_Turb_M1.0_Eta1e-08_Zeta1e-08, $M = 1.0$) serves as the primary testbed for capturing shock waves and high-Reynolds dynamics. PDEBench remains an actively used benchmark in recent scientific ML work on PDE modeling (Chen et al., 2025; Hu et al., 2025), partly because dense spatiotemporal ground truth is rarely available in real physical systems. 2) **Urban Traffic (TaxiBJ)** (Zhang et al., 2017): To verify transferability, we model macroscopic crowd flow as a pseudo-fluid, testing the G-KSCH basis on discrete latent advection-diffusion dynamics. 3) **Global Weather (WeatherBench)** (Rasp et al., 2020): We employ ERA5 reanalysis data (Hersbach et al., 2020) to stress-test the model on chaotic atmospheric thermodynamics and Coriolis effects, representing global-scale systems with ill-defined governing equations.

### 4.1.2. BASELINES

We compare APIC against a comprehensive suite of SOTAs across four categories: 1) **Physics-agnostic Data-Driven:** ResNet-3D (Hara et al., 2018) and SimpleCNN-3D (LeCun et al., 1998). 2) **Neural Operators:** FNO-3D (Li et al., 2021), FactFormer (Li et al., 2023), GNOT (Hao et al., 2023), UNO (Rahman et al., 2022), and a memory-optimized Transolver (Patched) for fair high-dimensional comparison (Wu et al., 2024). 3) **Physics-Informed:** PeRCNN (Rao et al., 2023) and PhyDNet. 4) **Cross-Domain:** SimVP (Gao et al., 2022), SimVP++ (Tan et al., 2022), PredRNN (Wang et al., 2017), ConvLSTM (Shi et al., 2015),

and FourCastNet (Pathak et al., 2022) for video and meteorological tasks.

### 4.1.3. TRAINING PROTOCOLS & EVALUATION METRICS

All models are implemented in PyTorch on a single NVIDIA RTX 6000 GPU to ensure fair comparison. We universally employ the AdamW optimizer (Loshchilov & Hutter, 2019), utilizing ReduceLROnPlateau (patience=3) for physical fields (CFD/Weather) and OneCycleLR (Smith & Topin, 2019) for traffic dynamics to optimize convergence stability.

For the primary Compressible CFD testbed, we use the standard Mean Squared Error (MSE) to align with baseline protocols (Takamoto et al., 2022). For the auxiliary Traffic and Weather tasks, we adopt the Relative $L_2$ Loss to prioritize physical structural fidelity over pixel-wise smoothing.

Validation is standardized across domains. For Urban Traffic and Global Weather we report Root Mean Square Error (RMSE). For the 3D Compressible Turbulence task we use a multi-dimensional SciML suite: the Relative $H^1$ Error $\mathcal{E}_{H^1} = \sqrt{(||u - \hat{u}||_2^2 + \beta||\nabla u - \nabla\hat{u}||_2^2)/(||u||_2^2 + \beta||\nabla u||_2^2)}$ which captures shock sharpness via spatial gradients, the vorticity error $\mathcal{E}_\omega = ||\nabla \times u - \nabla \times \hat{u}||_2/||\nabla \times u||_2$ for small-scale turbulent details, and the Mass Conservation Residual $\mathcal{R}_{mass} = N^{-1} \sum ||\partial_t \rho + \nabla \cdot (\rho\mathbf{u})||_2^2$. CFD inputs use the pre-standardized PDEBench fields directly; Meteorological and Urban Traffic inputs are standard-normalized.

## 4.2. Experiment I: Compressible Fluid Dynamics

### 4.2.1. QUANTITATIVE COMPARISON

As shown in Table 1, across all data splits, APIC consistently achieves SOTA relative $L_2$ performance. Most notably, in the extreme 5% low-data regime, APIC (0.059) significantly outperforms Neural Operators like 3D-FNO (0.145) and establishes a performance level comparable to ResNet-3D trained on 20% of the data (0.058). This confirms that physical priors effectively compress the hypothesis space, substantially enhancing Data Efficiency.

Our multi-dimensional evaluation exposes the structural limitations of physics-agnostic baselines. While SimpleCNN and ResNet-3D yield competitive $L_2$ errors, their significantly higher $H^1$ and Vorticity metrics reveal the "$L_2$ Trap": minimizing pixel-wise error by generating overly smooth fields that obliterate high-frequency turbulent structures (Rahaman et al., 2019). Conversely, Transolver exhibits the lowest Mass Error (0.79–0.86), which, in the context of its high $L_2$ and $H^1$ errors ($> 0.5$), indicates a tendency toward "Mean Collapse"—achieving spurious conservation by predicting trivial, low-gradient fields. In contrast, APIC is the unique architecture that simultaneously maintains leading $H^1$ accuracy and physical consistency, proving it adheres

*Table 1.* Quantitative performance comparison under varying training set proportions. The training data consists of the initial 80%, 20%, 10%, or 5% segments of the total temporal sequence, while the final 20% of the sequence is consistently reserved as a fixed validation set to evaluate long-term predictive stability and prevent data leakage.

| Model | Params | VRAM(MB) | 80% | | | | 20% | | | | 10% | | | | 5% | | | |
|---|---|---|---|---|---|---|---|---|---|---|---|---|---|---|---|---|---|---|
| | | | $L_2\downarrow$ | $H_1\downarrow$ | Vort$\downarrow$ | Mass$\downarrow$ | $L_2\downarrow$ | $H_1\downarrow$ | Vort$\downarrow$ | Mass$\downarrow$ | $L_2\downarrow$ | $H_1\downarrow$ | Vort$\downarrow$ | Mass$\downarrow$ | $L_2\downarrow$ | $H_1\downarrow$ | Vort$\downarrow$ | Mass$\downarrow$ |
| ResNet-3D | 1.78M | 22713 | 0.043 | 0.092 | 0.130 | 5.65 | 0.058 | 0.122 | 0.173 | 5.52 | 0.060 | 0.144 | 0.201 | 5.93 | 0.071 | 0.175 | 0.246 | 5.12 |
| SimpleCNN | 0.34M | 5364 | 0.073 | 0.150 | 0.212 | 5.39 | 0.085 | 0.171 | 0.239 | 5.40 | 0.082 | 0.167 | 0.235 | 5.35 | 0.092 | 0.189 | 0.265 | 5.20 |
| ConvLSTM | 0.48M | 8465 | 0.198 | 0.378 | 0.484 | 3.83 | 0.203 | 0.388 | 0.494 | 3.60 | 0.208 | 0.397 | 0.506 | 3.50 | 0.215 | 0.411 | 0.523 | 3.29 |
| 3D-FNO | 3.28M | 4879 | 0.112 | 0.288 | 0.422 | 4.83 | 0.121 | 0.306 | 0.446 | 4.73 | 0.131 | 0.321 | 0.464 | 4.70 | 0.145 | 0.344 | 0.490 | 4.54 |
| FactFormer | 0.33M | 91119 | 0.095 | 0.196 | 0.266 | 4.58 | 0.216 | 0.448 | 0.564 | 2.79 | 0.282 | 0.551 | 0.645 | 1.30 | 0.292 | 0.561 | 0.650 | 0.99 |
| PDEBench | 1.36M | 4074 | 0.067 | 0.148 | 0.215 | 5.38 | 0.080 | 0.176 | 0.253 | 5.21 | 0.091 | 0.199 | 0.282 | 5.36 | 0.100 | 0.220 | 0.316 | 5.05 |
| Transolver | 0.32M | 87462 | 0.296 | 0.568 | 0.658 | **0.86** | 0.296 | 0.568 | 0.658 | **0.84** | 0.296 | 0.568 | 0.658 | **0.84** | 0.297 | 0.569 | 0.659 | **0.79** |
| Transolver$_{Patched}$ | 36.85M | 6096 | 0.121 | 0.253 | 0.349 | 4.90 | 0.145 | 0.308 | 0.428 | 4.53 | 0.152 | 0.324 | 0.449 | 4.43 | 0.158 | 0.339 | 0.470 | 4.46 |
| UNO | 6.82M | 4833 | 0.111 | 0.286 | 0.419 | 4.85 | 0.122 | 0.305 | 0.443 | 4.80 | 0.132 | 0.321 | 0.462 | 4.76 | 0.146 | 0.344 | 0.487 | 4.62 |
| PhyDNet | 1.23M | 12506 | 0.089 | 0.190 | 0.267 | 5.40 | 0.098 | 0.208 | 0.292 | 5.16 | 0.103 | 0.226 | 0.315 | 4.95 | 0.112 | 0.244 | 0.338 | 4.92 |
| PeRCNN | 0.13M | 10522 | 0.123 | 0.262 | 0.365 | 4.79 | 0.144 | 0.300 | 0.407 | 4.33 | 0.157 | 0.325 | 0.438 | 4.07 | 0.167 | 0.343 | 0.455 | 4.21 |
| APIC(ours) | 1.36M | 21976 | **0.033** | **0.072** | **0.106** | 5.62 | **0.042** | **0.092** | **0.134** | 5.58 | **0.051** | **0.108** | **0.158** | 5.51 | **0.059** | **0.125** | **0.182** | 5.50 |

to conservation laws without sacrificing the sharpness of high-frequency discontinuities.

### 4.2.2. VISUALIZATION ANALYSIS

As illustrated in Figure 3, APIC demonstrates superior spatial reconstruction fidelity across all planar slices. In the medium-data regime, standard baselines exhibit characteristic spectral smoothing at shock interfaces, failing to resolve sharp discontinuities. Conversely, APIC precisely reconstructs high-gradient shock fronts with visual acuity comparable to the ground truth. This performance advantage becomes particularly pronounced in the data-scarce regime, where baselines suffer from mean collapse or high-frequency artifacts, while APIC preserves sharp interfaces. The error maps confirm that APIC's residuals remain small across the field, whereas other methods show concentrated high-error regions near shock fronts. See Appendix D for extended visualizations across all regimes.

### 4.2.3. ABLATION EXPERIMENTS

To deconstruct the performance sources of APIC, we conducted a series of ablation studies:

*Table 2.* Ablation across data splits ($L_2$, lower is better). Best in **bold**, second-best underlined.

| Model | 80% | 20% | 10% | 5% |
|---|---|---|---|---|
| Physics-agnostic Data-driven | 0.0530 | 0.0678 | 0.0807 | 0.0865 |
| Gated with NS | 0.0447 | 0.0586 | 0.0673 | 0.0794 |
| Stacked with NS | 0.0377 | 0.0481 | 0.0522 | 0.0646 |
| Stacked with G-KSCH | **0.0331** | **0.0425** | **0.0505** | **0.0586** |

Table 2 validates the architectural design layer-by-layer. First, Gated-Fusion with NS outperforms the data-driven

baseline, proving the efficacy of the dual-branch physics-aware structure. Second, *Stacked-Fusion* surpasses linear gating, confirming its superiority in capturing non-linear dynamics; while this incurs overhead about 0.45M params, 5GB VRAM, it yields substantial accuracy gains. Finally, the performance leap with *G-KSCH* underscores that operator completeness provides the critical inductive bias essential for generalization.

### 4.2.4. STREAM ISOLATION AND GRADIENT DETACH ANALYSIS

To investigate whether the two streams are individually sufficient and to quantify the role of the stop-gradient operation, we conducted a controlled ablation comparing four configurations: **DataOnly** (data backbone only), **PurePhysics** (physics branch with learnable $\lambda$ only), **APIC (no-detach)** (full architecture without stop-gradient), and **APIC (detach)** (the proposed gradient-isolated design).

*Table 3.* Stream isolation & training-strategy ablation on 3D compressible turbulence ($L_2$, lower is better). All values are derived from an independently retrained batch. Best per column **bolded**.

| Configuration | 10% | 5% |
|---|---|---|
| DataOnly | 0.078 | 0.093 |
| PurePhysics | 0.319 | 0.334 |
| Stage-wise + 3-layer adapter | 0.103 | 0.114 |
| APIC (no-detach) | **0.050** | **0.060** |
| APIC (detach, ours) | **0.050** | 0.062 |

Table 3 yields four observations. (i) Physics-only severely underfits ($L_2 > 0.31$): the operator basis alone cannot match observation-level accuracy without a data-driven residual corrector. (ii) Data-only is consistently weaker than the full APIC, showing the physics branch contributes

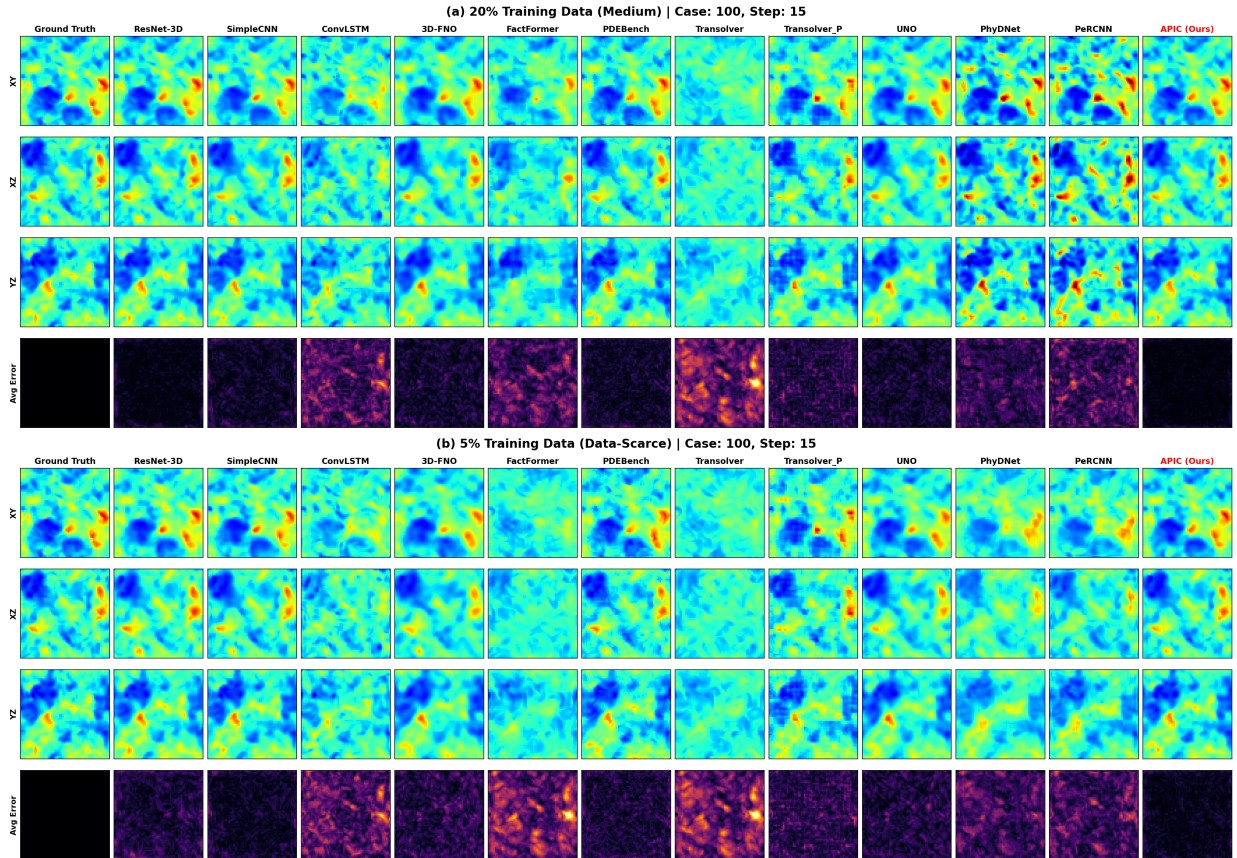

*Figure 3.* **Visualization of Prediction Results.** Comparison of spatial density fields during the critical moment of shock collision. **(a)** Predictions trained on 20% of the data sequence. **(b)** Predictions in the extreme 5% low-data regime.

substantively even when separately trainable. (iii) The stage-wise variant—training the two streams in isolation, freezing them, and then fitting only a 3-layer 3D adapter on top— collapses to $L_2 = 0.114$ at the 5% split, worse even than DataOnly. This confirms that joint training is necessary: the two branches must *co-adapt* through the shared prediction target, because frozen physics features and frozen data features occupy mismatched semantic manifolds that a small adapter cannot reconcile. (iv) The no-detach and detach variants achieve nearly identical $L_2$ accuracy; detach therefore improves *training stability and role separation* more than raw accuracy. Without detach, the learned $\lambda$ coefficients tend to be driven by end-to-end residual fitting, losing their role as structured physical references. With detach, the physics branch maintains its intended function—providing a structured prior that constrains the hypothesis space—while the data backbone retains full flexibility to learn whatever the physics cannot capture.

**Observation Noise.** Since the physics branch relies on finite-difference (FDM) stencils, we additionally evaluate robustness to i.i.d. Gaussian input noise at $1\%/5\%/10\%$ of the field magnitude. APIC degrades gracefully and consistently retains a substantial margin over the DataOnly baseline at every noise level, indicating that the G-KSCH inductive bias outweighs the noise penalty inherited from FDM. Full per-noise-level results are provided in Appendix D.1.5.

### 4.3. Experiment II: Cross-Task Adaptability

Rather than pursuing domain-specific SOTA, this section scrutinizes APIC's cross-task adaptability and Parameter Efficiency within non-traditional physical systems.

#### 4.3.1. TRAFFIC FORECASTING RESULTS

*Table 4.* Experimental results on Traffic dataset. Best results are **bolded**, and second-best results are underlined.

| Model | Params | RMSE↓ | MAE↓ | MAPE (%)↓ |
|---|---|---|---|---|
| PredRNN | 13.76M | 25.475 | 14.785 | 21.492 |
| ConvLSTM | 1.04M | 27.002 | 16.317 | 27.870 |
| PhyDNet | 0.13M | 36.531 | 23.123 | 40.922 |
| SimVP | 27.14M | 19.172 | 11.659 | 18.291 |
| SimVP_v2 | 5.45M | 18.355 | 11.576 | 18.943 |
| APIC | 0.46M | **17.887** | **11.005** | **17.677** |

For the Traffic dataset, we qualitatively adapted the operator basis to match crowd dynamics: amplifying the reaction

term (stop-and-go), removing hyper-diffusion, and employing a Single-well Potential for congestion. Consequently, despite possessing merely $0.46$M parameters, APIC outperforms the parameter-heavy SimVP_v2 ($5.45$M) in RMSE. This efficiency confirms that precise *Structural Adaptation* of physical priors effectively compresses the hypothesis space, enabling strong performance with far fewer parameters than purely data-driven alternatives.

To further verify that each adapted operator carries nontrivial weight, we conduct a component-wise ablation by disabling individual terms in the Traffic-customized G-KSCH basis.

*Table 5.* Operator-component ablation on Traffic (TaxiBJ). All variants in this ablation are evaluated in an independent run. Best results are **bolded**.

| Variant | RMSE ↓ | MAE ↓ | MAPE (%) ↓ |
|---|---|---|---|
| Full APIC (Traffic) | **17.810** | **10.884** | **17.14** |
| w/o Reaction term | 18.097 | 11.023 | 17.45 |
| w/o Congestion potential | 17.959 | 10.919 | 17.15 |
| Phase-like only (no transport) | 17.923 | 10.963 | 17.30 |

The component ablation in Table 5 shows that each adapted operator contributes measurably: removing the reaction term raises RMSE the most (to 18.097), while the congestion potential and the advection-dominant configuration each have a smaller but consistent effect. The gradual rather than catastrophic degradation indicates that APIC does not rely on a single brittle hand-crafted term, and that the structural reconfiguration is task-meaningful rather than cosmetic.

### 4.3.2. QUANTITATIVE RESULTS AND PARAMETER EFFICIENCY

*Table 6.* Experimental results on Weather dataset. The RMSE values are converted to meters (m) by dividing by $g \approx 9.80665$. Best results are **bolded**, and second-best results are underlined.

| Model | Params | RMSE (m) ↓ |
|---|---|---|
| APIC | 1.35M | 16.8842 |
| ConvLSTM | 1.15M | 29.2633 |
| FourCastNet | 3.43M | 22.6109 |
| PhyDNet | 1.31M | 40.0366 |
| PredRNN | 22.18M | **14.8933** |
| SimVP_v2 | 0.70M | 31.9627 |

For meteorological dynamics dominated by Anisotropy and Coriolis Effects, we adapted the operator basis by introducing a rotation term and decoupling horizontal/vertical coefficients. This structural adaptation alone reduced RMSE from $18.26$ to $16.88$. While marginally trailing PredRNN in accuracy, APIC achieves comparable performance using only $6\%$ of the parameter count, highlighting its critical value for deployment in resource-constrained environments.

## 5. Conclusion and Future Work

In this paper, we propose the APIC meta-architecture, which effectively reduces gradient interference between physical constraints and data-driven optimization via a gradient-isolated neuro-symbolic fusion strategy. Experiments demonstrate that APIC not only achieves SOTA performance in low-data regimes for compressible fluid dynamics but also exhibits strong cross-task adaptability through structural operator reconfiguration. When the backbone is not overfitting, richer physical priors consistently improve parameter efficiency. This also suggests a diagnostic use: by fixing the neural backbone and swapping operator combinations, one can observe how prediction loss shifts—providing a lightweight probe for unknown governing dynamics.

Currently, APIC is primarily applicable to Euclidean grid data and dissipative systems. Future work will focus on several directions: (i) integrating Graph Neural Networks to extend APIC to non-Euclidean geometries and unstructured meshes; (ii) exploring Hamiltonian or Lagrangian operator bases suitable for non-dissipative systems, thereby encompassing a broader range of field-theoretic physical problems; (iii) extending the gradient-isolation framework beyond PDE settings to ODE/SDE-governed dynamics, where the operator basis can be reformulated in terms of vector fields and stochastic drift–diffusion terms; (iv) applying post-hoc interpretability tools such as SHAP-style attribution to the learned adaptive coefficients $\lambda$, in order to quantitatively characterize which physical terms dominate the prediction in different regimes; and (v) extending APIC to action-conditioned dynamics and larger-scale engineering scenarios such as robotics simulation or real-world CFD pipelines, so that the framework can move beyond pure state forecasting toward closed-loop control settings.

## Impact Statement

We discuss the potential positive impacts and limitations of this work along the following dimensions.

**Potential positive impacts.** (i) For scientific forecasting tasks, APIC may improve data efficiency and physical consistency, and may also reduce training cost to some extent. (ii) For scientific modeling more broadly, the explicit operator basis and the learnable adaptive coefficients $\lambda$ may make hybrid forecasting models easier to inspect than fully black-box alternatives. In particular, this opens the possibility of using APIC as a form of *scientific probe*: researchers may replace the operator basis and observe how predictions and identified coefficients shift, providing a lightweight mechanism for partial scientific validation.

**Potential limitations and risks.** (i) If the chosen operator basis is fundamentally mismatched to the target system, the structured prior may guide the model in an unhelpful

direction rather than improve it. (ii) Strong benchmark performance should not be over-interpreted as full reliability in real scientific deployment, especially in safety-critical settings such as numerical weather prediction or infrastructure monitoring; appropriate validation against ground-truth simulations and domain expert review remain essential before deployment. (iii) For scientific tasks, human judgment remains important when interpreting predictions, particularly under distribution shift or limited observations; APIC is intended to assist—not replace—domain experts in such regimes.

The datasets used in this work are publicly available simulation benchmarks; we do not foresee deployment risks beyond those inherent to benchmark-driven scientific ML research at this stage.

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

# A. Derivation of the G-KSCH Operator Basis

In this section, we provide the complete mathematical derivation of the Generalized Kuramoto-Sivashinsky-Cahn-Hilliard (G-KSCH) operator. We demonstrate how this unified basis is synthesized from the Reynolds Transport Theorem (RTT) and non-equilibrium thermodynamics, bridging the gap between macroscopic fluid mechanics and microscopic phase separation dynamics.

## A.1. Foundation: Reynolds Transport Theorem and Conservation Laws

The derivation begins with the fundamental conservation principles governing a dynamic field $u$ within a control volume $\Omega(t)$. The Reynolds Transport Theorem (RTT) relates the rate of change of an extensive property within a moving control volume to the external flux crossing its boundaries. Let $u$ be the intensive property (quantity per unit volume) corresponding to the field. The RTT states:

$$\frac{d}{dt} \int_{\Omega(t)} u \, dV = \int_{\Omega(t)} \frac{\partial u}{\partial t} \, dV + \int_{\partial \Omega(t)} u(\mathbf{v}_b \cdot \mathbf{n}) \, dA \tag{5}$$

In an Eulerian framework with a fixed grid, the boundary velocity is zero ($\mathbf{v}_b = 0$). According to the general conservation law, the rate of change of the total quantity in the volume equals the sum of the source generation $S(u)$ and the net flux $\mathbf{J}$ crossing the boundary:

$$\int_{\Omega} \frac{\partial u}{\partial t} \, dV = -\int_{\partial \Omega} \mathbf{J} \cdot \mathbf{n} \, dA + \int_{\Omega} S(u) \, dV \tag{6}$$

Applying Gauss's Divergence Theorem to convert the boundary flux into a volume integral:

$$\int_{\Omega} \left( \frac{\partial u}{\partial t} + \nabla \cdot \mathbf{J} - S(u) \right) dV = 0 \tag{7}$$

Since this holds for any arbitrary control volume $\Omega$, the integrand must vanish, yielding the fundamental Generalized Flux Differential Form used in APIC:

$$\frac{\partial u}{\partial t} + \nabla \cdot \mathbf{J}(u) = S(u) \tag{8}$$

Based on the Generalized Flux Decomposition proposed in the paper, we assert that the diverse behaviors of dissipative systems arise solely from differences in the constitutive relations of the flux $\mathbf{J}$.

## A.2. Component Derivations

We now derive the specific mathematical form for each flux component and its contribution to the time evolution $\partial_t u$.

### A.2.1. NONLINEAR ADVECTION (MACROSCOPIC TRANSPORT)

Advection describes the transport of a quantity $u$ by a flow field. In the context of the Burgers equation or momentum transport, the flux is convective.

- **Flux Form:** $\mathbf{J}_{adv} = \frac{1}{2}u^2$ (scalar) or $u \otimes u$ (vector).
- **Derivation:** The divergence of the convective flux corresponds to the material derivative term:

$$\nabla \cdot \mathbf{J}_{adv} = \nabla \cdot \left( \frac{1}{2}u^2 \right) = u \cdot \nabla u \tag{9}$$

- **Result:** Assuming a quasi-incompressible phenomenological prior for the operator basis, this yields the nonlinear advection term, describing the macroscopic migration of matter:

$$\text{Term}_{adv} = -\lambda_1(u \cdot \nabla u) \tag{10}$$

### A.2.2. FICKIAN DIFFUSION (MICROSCOPIC DISSIPATION)

Diffusion arises from random Brownian motion tending to smooth out gradients, governed by Fick's First Law.

- **Flux Form:** $\mathbf{J}_{diff} = -\nu \nabla u$, where $\nu$ is the diffusion coefficient.

- **Derivation:** Substituting this flux into the conservation equation:

$$-\nabla \cdot \mathbf{J}_{diff} = -\nabla \cdot (-\nu \nabla u) = \nu \nabla^2 u \tag{11}$$

- **Result:** This provides the second-order Laplacian smoothing term:

$$\text{Term}_{diff} = \lambda_2 \nabla^2 u \tag{12}$$

### A.2.3. PHASE SEPARATION (GINZBURG-LANDAU POTENTIAL)

To capture non-linear pattern formation, we incorporate phase separation dynamics derived from the Cahn-Hilliard framework.

- **Chemical Potential:** The chemical potential $\mu$ is the variational derivative of the Ginzburg-Landau free energy $F[u]$. Focusing on the bulk potential part:

$$\mu \approx \frac{\delta F}{\delta u} \propto u^3 - u \tag{13}$$

- **Flux Form:** The flux is driven down the gradient of the chemical potential: $\mathbf{J}_{sep} = -\nabla \mu = -\nabla(u^3 - u)$.
- **Derivation:** Taking the divergence:

$$-\nabla \cdot \mathbf{J}_{sep} = \nabla^2(u^3 - u) \tag{14}$$

- **Result:** This introduces a "reverse diffusion" mechanism, forcing the system to separate into sharp phases:

$$\text{Term}_{sep} = \lambda_3 \nabla^2(u^3 - u) \tag{15}$$

### A.2.4. HYPER-DIFFUSION (FOURTH-ORDER REGULARIZATION)

The fourth-order term is necessary to stabilize the ill-posedness introduced by the phase separation term.

- **Derivation:** Originating from the gradient penalty in the free energy, the flux corresponds to $\mathbf{J}_{hyper} = \nabla(\nabla^2 u)$.

$$-\nabla \cdot \mathbf{J}_{hyper} = -\nabla \cdot \nabla(\nabla^2 u) = -\nabla^4 u \tag{16}$$

- **Result:** This acts as a fourth-order regularization:

$$\text{Term}_{hyper} = -\lambda_4 \nabla^4 u \tag{17}$$

## A.3. Synthesis: The G-KSCH Meta-Equation

Combining all derived generalized flux divergences and the source term $S(u) = \lambda_5 u$ into the governing equation, we arrive at the unified G-KSCH operator basis:

$$\frac{\partial u}{\partial t} = \underbrace{-\lambda_1(u \cdot \nabla u)}_{\text{Advection}} + \underbrace{\lambda_2 \nabla^2 u}_{\text{Diffusion}} + \underbrace{\lambda_3 \nabla^2(u^3 - u)}_{\text{Phase Separation}} - \underbrace{\lambda_4 \nabla^4 u}_{\text{Hyper-diffusion}} + \underbrace{\lambda_5 u}_{\text{Reaction}} \tag{18}$$

where $\lambda_i$ are adaptive coefficients learned via the Gradient Isolation mechanism.

## A.4. Task-Specific Operator Adaptations

While the G-KSCH basis provides a universal foundation, APIC leverages *Structural Isomorphism* to adapt to specific domain priors. Below we detail the operator reconfigurations for the Traffic and Meteorology tasks.

### A.4.1. URBAN TRAFFIC FLOW (CONGESTION DYNAMICS)

In traffic modeling, the field variable $u$ represents vehicle density $\rho$. Based on the LWR model and congestion mechanics, we reconfigure the operators as follows:

- **Bounded Advection:** To model speed limits and saturation, we introduce a $\tanh$ activation:

$$\mathcal{F}_{adv} = -\lambda_1 \tanh(\rho)|\nabla \rho| \tag{19}$$

where $\lambda_1 = 1 + 0.1 \tanh(\alpha_{adv})$ ensures advection remains the dominant mechanism.

- **Congestion Potential (Nonlinear Diffusion):** We replace the phase separation term with a Congestion Potential based on squared density, $P(\rho) \propto \rho^2$. The flux divergence becomes:

$$\mathcal{F}_{cong} = \lambda_3 \nabla^2(\rho^2) \tag{20}$$

  **Physical Interpretation:** This creates a density-dependent diffusion rate. As traffic density $\rho$ increases, the effective diffusion coefficient scales with $2\rho$, simulating the "repulsive" behavior where vehicles rapidly evacuate from high-density congestion centers.

- **Enhanced Reaction:** The source term coefficient $\lambda_5$ is expanded to the range $[-0.5, 0.5]$ to capture strong source/sink dynamics (e.g., parking lots, intersections) characteristic of urban traffic.

### A.4.2. GLOBAL METEOROLOGY (ANISOTROPIC & ROTATIONAL DYNAMICS)

Meteorological data exhibits significant anisotropy due to atmospheric stratification and the Coriolis effect.

- **Anisotropic Decomposition:** We decompose the isotropic Laplacian into horizontal ($\nabla_h^2$) and vertical ($\nabla_v^2$) components with independent adaptive coefficients:

$$\lambda_2 \nabla^2 u \longrightarrow \lambda_{2h} \nabla_h^2 u + \lambda_{2v} \nabla_v^2 u \tag{21}$$

  Implementation sets horizontal transport as dominant ($\lambda_{2h} \approx 1.0$) while vertical exchange is treated as a perturbation ($\lambda_{2v} \approx 0.01$) to reflect gravity-induced stratification.

- **Coriolis Rotation:** To implicitly account for asymmetric rotational dynamics induced by the Coriolis effect, we introduce a phenomenological rotational operator acting on the horizontal gradients:

$$\mathcal{F}_{rot} = \lambda_{rot} \left( \frac{\partial u}{\partial y} - \frac{\partial u}{\partial x} \right) \tag{22}$$

- **Stability Phase:** The high-order hyper-diffusion term is simplified to a linear anisotropic diffusion combination ($\lambda \approx 0.01$) to maintain numerical stability over long integration horizons without inducing high-frequency artifacts.

## B. Why `detach` Inside the Operator Argument: A First- and Second-Order Analysis

This appendix provides the formal justification for the precise placement of the `detach` operation in APIC, expanding on the brief remark in the *Gradient Isolation and Physics Adapter* subsection of the main text. The discussion is organised around three observations: (i) where on the computational graph the truncation lives, (ii) how it modifies the first-order Jacobian and thus prevents "gradient hijacking", and (iii) how it tames the second-order Hessian and stabilises optimisation.

### B.1. Topological Placement on the Computation Graph

In the APIC forward pass we have two parallel branches:

- **Data Stream**: a high-dimensional feature tensor produced by the neural backbone $\mathcal{N}_\theta(X)$.

- **Physics Stream**: a fixed-form operator basis $\mathcal{P}(\cdot)$ (the G-KSCH operator) that maps features through one step of physical evolution.

The fused output is

$$Y_{\text{out}} = \mathcal{N}_\theta(X) + \mathcal{P}\big(\texttt{detach}(\mathcal{N}_\theta(X))\big). \tag{23}$$

The exact location of `detach` matters:

- **Outside the operator**, $\texttt{detach}(\mathcal{P}(\mathcal{N}_\theta(X)))$: the truncation is *ineffective*. The forward pass is unchanged, but during the backward pass the operator's gradient still flows back through $\mathcal{N}_\theta$ and contaminates $\theta$.

- **Wholesale truncation** of both branches: the network parameters receive no useful gradient signal and training collapses.

- **Precisely on the bridge**, $\mathcal{P}(\texttt{detach}(\mathcal{N}_\theta(X)))$: the forward signal is delivered into the operator faithfully, but the backward edge through the operator is severed. Gradients can only flow along the data stream, yielding a one-way "semi-permeable membrane".

### B.2. First-Order Jacobian: Preventing Gradient Hijacking

Let $\mathcal{L}(Y_{\text{out}}, Y_{\text{true}})$ denote the prediction loss.

**Without `detach`.** By the chain rule,

$$\frac{\partial \mathcal{L}}{\partial \theta} = \frac{\partial \mathcal{L}}{\partial Y_{\text{out}}} \left[ \frac{\partial \mathcal{N}_\theta}{\partial \theta} + \underbrace{\frac{\partial \mathcal{P}}{\partial \mathcal{N}_\theta}}_{\text{operator Jacobian}} \cdot \frac{\partial \mathcal{N}_\theta}{\partial \theta} \right]. \tag{24}$$

The operator Jacobian $\partial \mathcal{P} / \partial \mathcal{N}_\theta$ is non-trivial. Because the backbone is locally greedy with respect to the loss, gradient descent can adjust $\theta$ so that the data-stream gradient *cancels* the operator-stream gradient. In effect, the network learns an internal "anti-operator" that nullifies the physics prior — a form of *gradient hijacking*. The recovered signal then no longer reflects a physically meaningful residual, but rather an arbitrary fit to the supervised target.

**With `detach`.** The detach operation enforces, by definition,

$$\frac{\partial \mathcal{P}}{\partial \mathcal{N}_\theta} \equiv \mathbf{0}. \tag{25}$$

The gradient update reduces to

$$\frac{\partial \mathcal{L}}{\partial \theta} = \frac{\partial \mathcal{L}}{\partial Y_{\text{out}}} \cdot \frac{\partial \mathcal{N}_\theta}{\partial \theta}. \tag{26}$$

Geometrically, the update direction for $\theta$ is stripped of the tangent components arising from the physics manifold. The data-stream gradient and the physics-stream constraint occupy non-interfering subspaces of the parameter space, and the backbone can only fit the residual unexplained by $\mathcal{P}$.

### B.3. Second-Order Hessian: Smoothing the Loss Landscape

A second perspective is curvature. The Hessian of the loss with respect to $\theta$ is, to leading order,

$$\mathbf{H} = \nabla_\theta^2 \mathcal{L} \approx (\nabla_\theta Y_{\text{out}})^\top \left( \nabla_{Y_{\text{out}}}^2 \mathcal{L} \right) (\nabla_\theta Y_{\text{out}}). \tag{27}$$

**Without `detach`.** The Jacobian $\nabla_\theta Y_{\text{out}}$ contains $\nabla_\theta \mathcal{P}$. Because $\mathcal{P}$ involves up to fourth-order spatial derivatives (see the G-KSCH definition in Section 3.2.2), it amplifies high-frequency components by factors of $\omega^k$ in the Fourier domain. The resulting Hessian has a severely degraded condition number: the loss surface develops sharp, high-curvature directions that destabilise first-order optimisers, often manifesting as oscillation or NaN losses.

**With `detach`.** The truncation zeroes out the high-order operator cross-terms in $\nabla_\theta Y_{\text{out}}$, so the corresponding contributions to $\mathbf{H}$ vanish. The forward physical manifold is preserved (predictions are still correct), but the high-frequency curvature induced by discretised differential operators is removed from the parameter-space view of $\mathcal{L}$. The optimiser sees a landscape dominated by the well-conditioned data-stream Jacobian, which is consistent with the empirically observed faster and more stable convergence reported in Appendix D.

### B.4. Summary

Placing detach *inside* the operator argument is therefore neither cosmetic nor merely engineering convenience. At first order it prevents the backbone from hijacking the physics prior; at second order it removes ill-conditioned high-frequency curvature introduced by the spatial differential operators. Together, these two effects justify the precise role of gradient isolation in APIC.

## C. Adaptive Coefficient Behaviour under Detach vs. No-Detach

To complement the loss-level ablation in Section 4.2.4, this appendix examines how the learned adaptive coefficients $\lambda$ of the G-KSCH operator behave across random seeds, with and without the gradient isolation (detach) operation. We ran both APIC variants under the same three-seed protocol ($\{43, 2019, 2026\}$) on the 5% and 10% data splits, and inspected the per-channel coefficients of the operator basis at convergence.

**Cross-seed variance is not, by itself, a reliable indicator.** A naive expectation is that `detach` should produce numerically smaller cross-seed standard deviations for $\lambda$. Empirically, this expectation is too coarse: on both the 10% (Mode-19) and 5% (Mode-0595) splits, the no-detach and detach variants can exhibit comparable or even reversed cross-seed standard deviations for some channels/operators. The reason is that a small variance may also arise when the coefficients collapse toward the boundary of their feasible range (saturating `tanh`/`sigmoid`) or settle into a degenerate compensation regime. Such "small-variance" configurations are not evidence of stable physics identification — they are evidence that the optimisation has been hijacked into a corner of the parameter space.

**What changes is the *role* of the coefficients.** Focusing on *role separation* rather than raw variance, a consistent qualitative pattern emerges. In the no-detach setting, several coefficients are pulled toward extreme values that are clearly driven by end-to-end residual fitting: the physics branch is partially repurposed to absorb errors from the data stream. The corresponding predictions remain competitive in $L_2$ (cf. Table 3 in the main text), but the identified $\lambda$ no longer admit a clean "physics-trend + data-residual" decomposition. By contrast, with `detach`, the physics coefficients are not directly driven by the residual gradient. They tend to settle into interior, non-degenerate values whose magnitudes are consistent with the physical role of the corresponding operator term (e.g. a dominant advection coefficient near unity, perturbative diffusion / hyper-diffusion at small magnitudes). This is the empirical counterpart of the first-order argument in Appendix B: removing the dominant indirect Jacobian path is what makes the coefficients reflect a stable trend-modelling role rather than a generic correction term.

**Implication for interpretability.** This is why, throughout the paper, we treat APIC's interpretability as coming primarily from its built-in structural design — the explicit operator basis and the role of $\lambda$ under `detach` — rather than from post-hoc tools alone. A complementary post-hoc analysis using SHAP-style attribution is discussed as a future direction in Section 5.

# D. Extended Experimental Results

In this section, we present a comprehensive evaluation of the APIC framework. Complementing the derivation in Appendix A, we provide detailed quantitative metrics and qualitative visualizations across Compressible Fluid Dynamics (CFD), Urban Traffic Forecasting, and Global Meteorological Prediction. These results empirically validate the theoretical advantages of the G-KSCH operator basis and the Gradient Isolation strategy.

## D.1. Compressible Fluid Dynamics (CFD)

The CFD experiment serves as the core testbed for verifying the model's ability to handle strong non-linearities (shock waves) and adhere to conservation laws in the raw physical space.

D.1.1. QUALITATIVE VISUALIZATION: SHOCK WAVE RECONSTRUCTION

We verify the model's capability to capture high-frequency turbulent structures under data-scarce conditions.

- **Visual Analysis:** Figure 4 compares the predictive fidelity across varying training data proportions ($80\% \rightarrow 5\%$).
- **Observation:** Even in the extreme $5\%$ low-data regime, APIC maintains sharp shock interfaces. In contrast, baseline Fourier Neural Operators (FNO) tend to exhibit ringing artifacts (Gibbs phenomenon) due to spectral bias, while standard CNNs produce overly smoothed fields lacking high-frequency details.

D.1.2. CONVERGENCE DYNAMICS & OPTIMIZATION STABILITY

We analyze the training stability provided by the Gradient Isolation strategy.

- **Loss Landscape:** Figure 5 illustrates the $L_2$ relative error trajectories.
- **Mechanism:** The decoupling of the physics parameter identification ($\lambda_i$) from the residual correction prevents the "tug-of-war" optimization pathology often observed in PINNs. This results in a steeper convergence rate and lower asymptotic error compared to coupled optimization baselines.

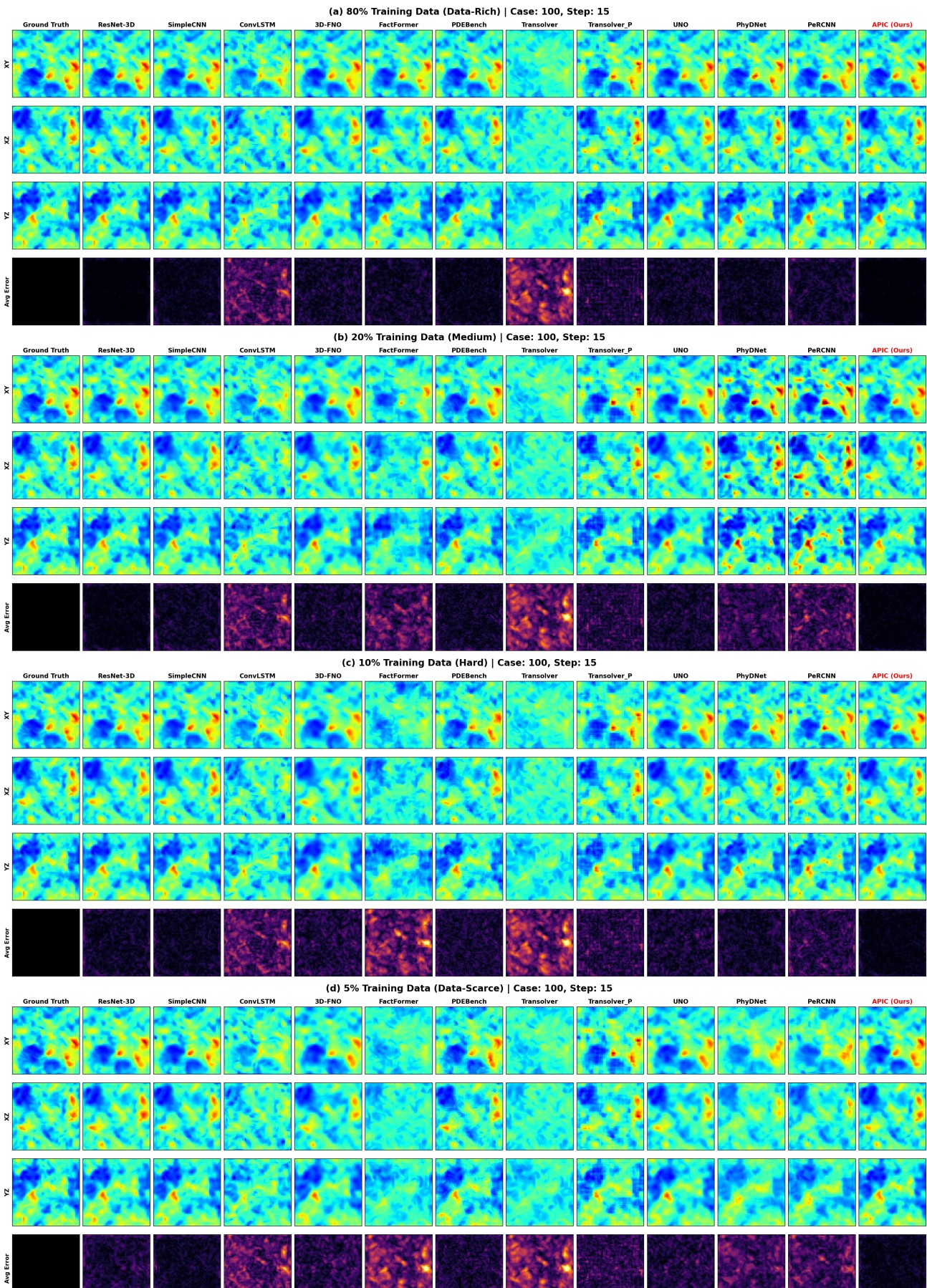

*Figure 4.* **Predictive Visualization across Data Regimes.** Comparison of ground truth vs. APIC predictions for $t + 10$ steps under 80%, 20%, 10%, and 5% training data splits. Note the preservation of sharp wavefronts in the 5% regime.

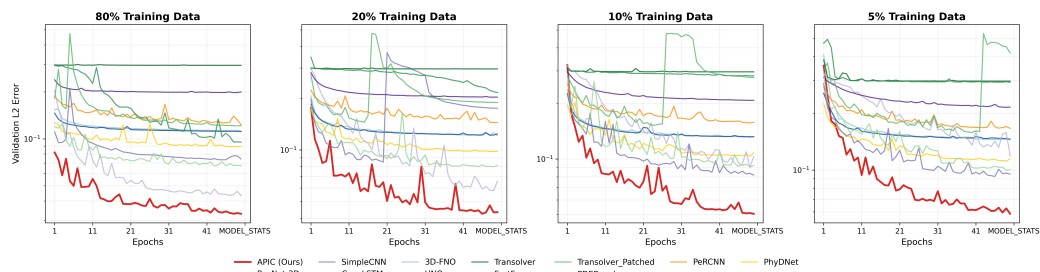

*Figure 5.* **Convergence Dynamics.** $L_2$ relative error curves during training. APIC demonstrates faster convergence and higher stability compared to end-to-end physics-constrained baselines.

### D.1.3. PARETO OPTIMALITY & COMPUTATIONAL EFFICIENCY

To demonstrate the practical viability of APIC, we conduct a multi-objective analysis involving accuracy, memory usage, and inference speed.

- **Pareto Frontier:** Figure 6 plots the Validation $L_2$ Error against computational resources (VRAM, Params, Time) and Physical Inconsistency (Mass Error).
- **Conclusion:** APIC (marked by the red star) consistently occupies the lower-left quadrant (Pareto optimal region). Specifically, in the 5% data-scarce regime, it achieves accuracy comparable to heavy Transformer models (e.g., Transolver) but with orders of magnitude lower latency and memory footprint.

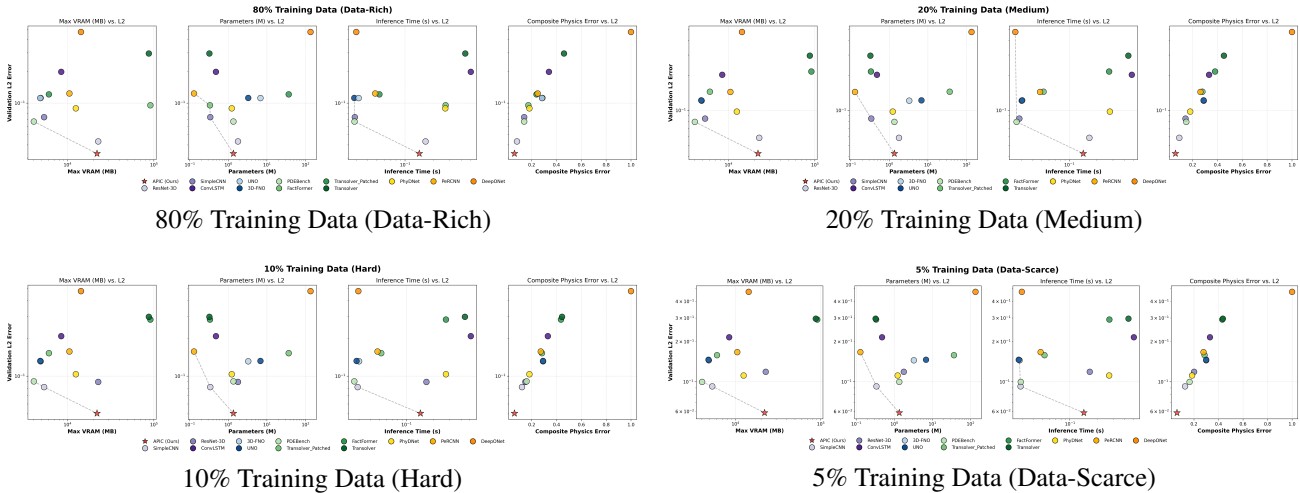

*Figure 6.* **Pareto Optimality Analysis.** Multi-dimensional efficiency comparison across varying data regimes. APIC defines the efficiency frontier, balancing high accuracy with low computational overhead.

### D.1.4. MULTI-SEED RELIABILITY ANALYSIS

To probe the statistical reliability of the main-table comparison, we re-ran the three most representative configurations—ResNet-3D (strongest data-driven baseline), 3D-FNO (representative neural operator), and APIC (ours)—under three random seeds ($\text{seed} \in \{43, 2019, 2026\}$) on the 5% and 10% data splits, which are the two regimes most relevant to our low-data claim. Other splits (20%, 80%) and other baselines were not re-seeded due to compute budget. Table 7 reports the resulting $L_2$ validation error per seed, the across-seed mean, and the resource footprint per run. All runs share the same training pipeline, optimizer, and stopping criterion used in the main paper.

Three observations follow. First, all three models are tightly clustered across seeds (std $< 0.006$ in every row), indicating that the cross-model ranking in the main table is not an artifact of a lucky seed. Second, APIC retains the lowest $L_2$ at every seed on both splits, including against the strongest seed of ResNet-3D (0.0701 vs. APIC 0.0597 at 5%; 0.0597 vs. APIC 0.0499 at 10%). Third, APIC's training cost per epoch is comparable to ResNet-3D and substantially below FactFormer

*Table 7.* Per-seed $L_2$ validation error and resource footprint for the three re-seeded models on the 5% and 10% splits. "Best" denotes the best single-seed result; "Mean" is the across-seed average. Main-table values follow the mean−std convention.

| Model | Split | seed 43 | seed 2019 | seed 2026 | Mean | Best | Train (s) | VRAM (MB) |
|---|---|---|---|---|---|---|---|---|
| ResNet-3D | 5% | 0.0701 | 0.0817 | 0.0788 | 0.0769 | 0.0701 | 37.2 | 22,753 |
| | 10% | 0.0597 | 0.0626 | 0.0664 | 0.0629 | 0.0597 | 74.0 | 22,753 |
| 3D-FNO | 5% | 0.1416 | 0.1429 | 0.1446 | 0.1430 | 0.1416 | 6.1 | 4,917 |
| | 10% | 0.1330 | 0.1306 | 0.1320 | 0.1319 | 0.1306 | 11.8 | 4,917 |
| APIC (ours) | 5% | 0.0597 | 0.0633 | 0.0617 | 0.0616 | 0.0597 | 30.2 | 21,976 |
| | 10% | 0.0504 | 0.0499 | 0.0502 | 0.0502 | 0.0499 | 60.1 | 21,976 |

/ Transolver (cf. Table 1), while its inference VRAM remains well within a single RTX 6000. These multi-seed results corroborate the data-efficiency claim and rule out seed-induced cherry-picking.

### D.1.5. ROBUSTNESS TO OBSERVATION NOISE

Since the physics branch of APIC computes spatial derivatives via fixed finite-difference (FDM) stencils, it is in principle sensitive to noisy observations. To quantify this, we inject i.i.d. Gaussian noise into the test inputs at standard deviations of 1%, 5%, and 10% of the field magnitude, and re-evaluate the trained models without any retraining or denoising.

*Table 8.* Noise robustness on 3D compressible turbulence ($L_2$ error under Gaussian input noise of varying magnitude). Results are from an independent evaluation run. Best per row in **bold**.

| Split | Model | 0% | 1% | 5% | 10% |
|---|---|---|---|---|---|
| 10% | DataOnly | 0.0834 | 0.0834 | 0.0845 | 0.0877 |
| | APIC (detach) | **0.0505** | **0.0505** | **0.0518** | **0.0560** |
| 5% | DataOnly | 0.1131 | 0.1132 | 0.1137 | 0.1154 |
| | APIC (detach) | **0.0619** | **0.0619** | **0.0630** | **0.0679** |

All configurations degrade gracefully as noise increases, consistent with the known sensitivity of FDM stencils. However, APIC consistently retains a substantial advantage over the DataOnly baseline at every noise level, and the relative gap widens as noise grows: at the 5% split with 10% noise, the gap reaches $\sim 41\%$ relative. The same trend holds under the $H^1$ metric (omitted for brevity). This indicates that while APIC inherits some sensitivity from its FDM-based physics branch, the inductive bias from the G-KSCH operator basis provides regularization that outweighs the noise penalty, yielding net robustness rather than fragility.

### D.1.6. 1D SHOCK WAVE EVALUATION

To further probe APIC's ability to handle extreme discontinuities, we additionally evaluate on the `1D_CFD_Shock_Eta1e-8_Zeta1e-8` dataset from PDEBench under the 5% data regime, using the same training pipeline as the 3D experiments. Results are averaged over 3 random seeds.

*Table 9.* 1D Shock dataset evaluation under the 5% data regime ($L_2$ error, mean $\pm$ std over 3 seeds).

| Model | $L_2 \downarrow$ |
|---|---|
| ResNet-3D | $0.0245 \pm 0.0070$ |
| 3D-FNO | $0.0150 \pm 0.0009$ |
| APIC (ours) | $\mathbf{0.0112 \pm 0.0007}$ |

APIC outperforms 3D-FNO by $\sim 25\%$ and ResNet-3D by over $50\%$ on this extreme-discontinuity benchmark, confirming that the G-KSCH operator basis transfers to lower-dimensional shock settings without architectural modification. This supplementary result, prepared in response to reviewer comments, complements the 3D compressible turbulence experiments in the main text.

## D.2. Urban Traffic Forecasting

This section validates the structural reconfigurability of APIC, specifically the adaptation of the G-KSCH kernel to model Congestion Potential and Source/Sink dynamics as derived in Appendix A.

### D.2.1. SPATIOTEMPORAL COHERENCE

- **Analysis:** Figure 7 visualizes the predicted traffic density evolution. The model successfully captures the non-linear propagation of congestion waves (red regions) and their dissipation, a behavior directly modeled by the adapted Phase Separation term $\nabla^2(\rho^2)$.

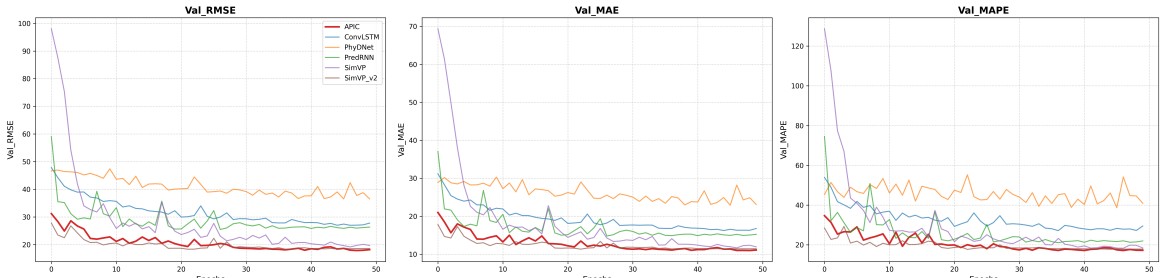

*Figure 7.* **Urban Traffic Visualization.** Prediction of macroscopic traffic flow density. APIC accurately tracks the formation and dispersion of high-density congestion clusters.

### D.2.2. METRIC EVOLUTION

- **Analysis:** Figure 8 shows the smooth decay of RMSE, MAE, and MAPE. The stability of these metrics confirms that the modified operator basis ($[-C, C]$ reaction range) effectively regularizes the learning process on discrete grid data.

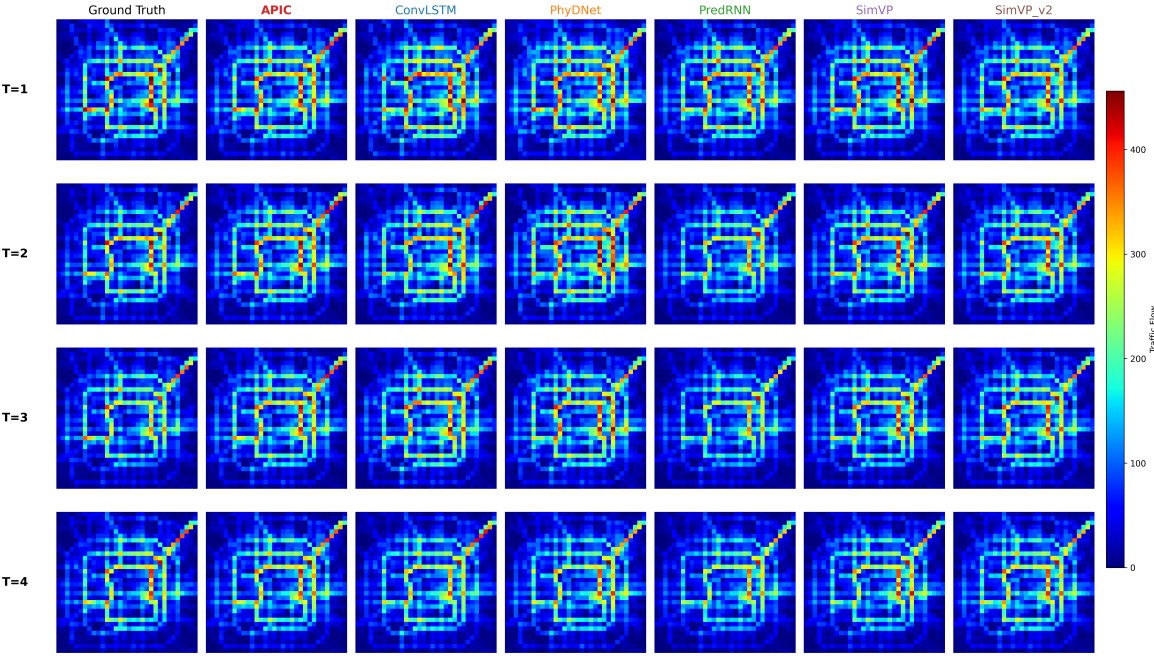

*Figure 8.* **Traffic Metrics Evolution.** Validation curves showing consistent error reduction, indicating robust parameter identification for the traffic-specific operators.

## D.3. Global Meteorology

Finally, we demonstrate APIC's generalization to spherical atmospheric dynamics, utilizing the anisotropic operator decomposition ($\nabla_h^2, \nabla_v^2$) and rotation terms.

### D.3.1. ANISOTROPIC FIELD RECONSTRUCTION

- **Analysis:** As shown in Figure 9, APIC captures the swirling structure of geopotential height fields. The inclusion of the Coriolis rotation term allows the model to correctly predict the rotational movement of high-pressure systems.

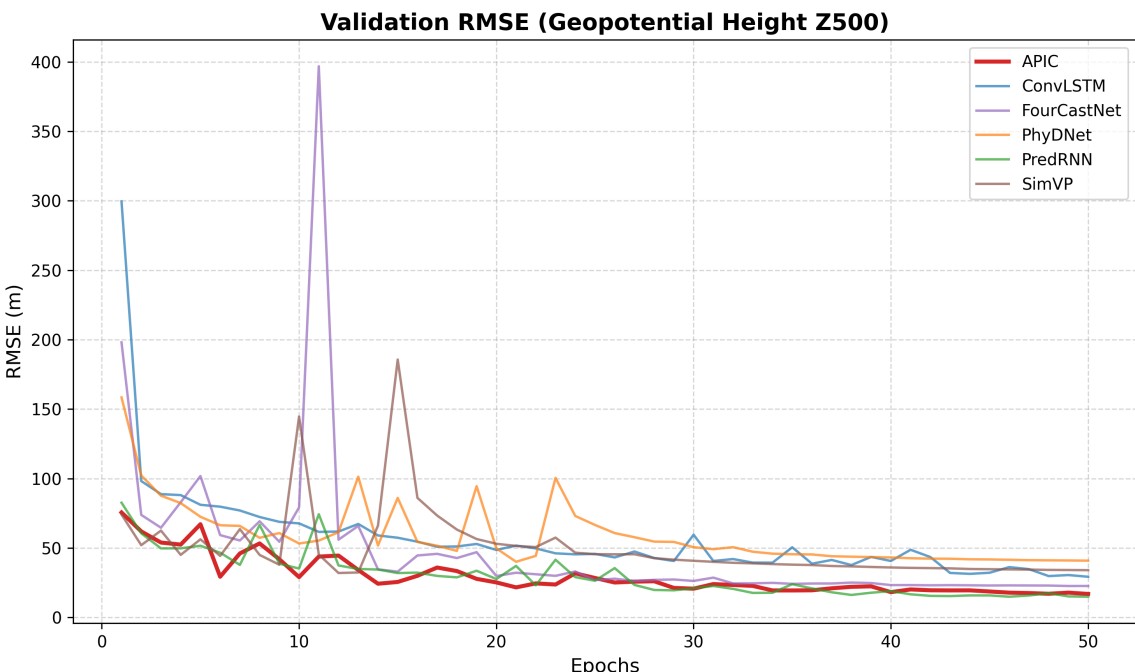

*Figure 9.* **Meteorological Field Prediction.** Visualization of Geopotential Height ($Z500$) and Temperature ($T850$). The model preserves the anisotropic structure of atmospheric waves.

### D.3.2. LONG-HORIZON STABILITY

- **Analysis:** Figure 10 displays the validation loss over extended horizons. Unlike pure data-driven models that often suffer from error accumulation (divergence), APIC maintains physical consistency (low mass/energy residual) due to the constraint of the underlying conservation laws.

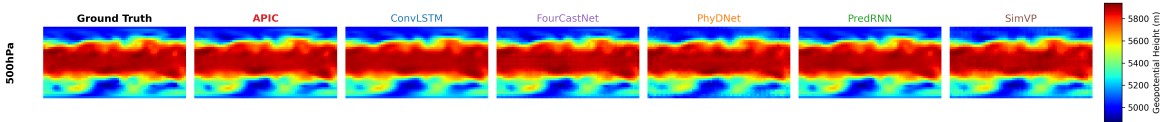

*Figure 10.* **Meteorological Validation Metrics.** The low asymptotic error validates the effectiveness of the Anisotropic Operator Decomposition in handling global-scale atmospheric dynamics.

