# OpenReview forum: "APIC: Orthogonalized Neuro-Symbolic Modeling for Nonlinear Dissipative Dynamics"
_ICML.cc/2026/Conference — ICML 2026 regular_

### Official Review · Reviewer_SuXR · 2026-03-09

**Soundness:** 2
**Presentation:** 2
**Significance:** 1
**Originality:** 2
**Overall Recommendation:** 3
**Confidence:** 3

**Summary:**

The paper proposes APIC (Adaptive Physics-Informed Computing), a neuro-symbolic framework for modeling nonlinear dissipative dynamics. The method introduces a dual-stream architecture that separates physics-based modeling from data-driven residual learning. A unified PDE operator (G-KSCH) is used in the physics stream to model dissipative dynamics, while a neural network learns residual corrections in the data stream. The two streams are combined through feature stacking and a gradient isolation mechanism designed to reduce optimization conflicts between physics parameter learning and neural residual modeling. Experiments on PDEBench, TaxiBJ, and WeatherBench show improvements in predictive accuracy and data efficiency compared with several neural operator and spatiotemporal forecasting baselines. However, the empirical evaluation is relatively limited and does not clearly demonstrate strong advantages in more realistic scientific modeling scenarios.

**Compliance With Llm Reviewing Policy:**

Affirmed.

**Final Justification:**

The additional results solved my concerns

**Key Questions For Authors:**

## Questions

1. The experimental evaluation focuses mainly on benchmark prediction tasks. Can the authors demonstrate the effectiveness of the proposed framework on more realistic scientific simulation problems or engineering systems?

2. How sensitive is the method to the specific design of the G-KSCH operator basis? Would the performance degrade significantly if certain terms were removed or modified?

3. The architecture introduces several additional components compared to standard neural operator models. What is the computational overhead in training and inference?

4. The proposed framework relies on manually constructed physical operators. How generalizable is this approach to domains where the governing equations are unknown or only partially understood?

5. The current formulation appears limited to dissipative PDE systems defined on regular grids. How could the method be extended to non-Euclidean domains, irregular meshes, or Hamiltonian systems?

6. The model predicts future states based solely on previous states. How could the framework be extended to handle action-conditioned dynamics required for control and simulation tasks?

**Limitations:**

yes

**Strengths And Weaknesses:**

## Strengths

**1. Clear architectural design**

The paper proposes a structured dual-stream architecture that explicitly separates physics-based modeling and data-driven residual learning. This design provides an intuitive interpretation of how physics priors and neural networks interact.

**2. Physically motivated operator design**

The proposed G-KSCH operator attempts to unify several dissipative processes within a single framework. This formulation introduces interpretable physical components such as advection, diffusion, and reaction terms.

**3. Attempt to address gradient conflicts in hybrid models**

The gradient isolation mechanism is intended to mitigate optimization conflicts between physics parameter identification and neural residual learning, which is a known issue in physics-informed learning frameworks.

**4. Demonstrated improvements on selected benchmarks**

The method shows improvements on certain evaluation metrics in the PDEBench turbulence dataset, particularly in low-data regimes.

## Weaknesses

**1. Experimental tasks remain relatively toy and limited**

Despite claims of modeling complex dynamics, the experiments are conducted primarily on standard benchmark prediction tasks. The setups resemble spatiotemporal forecasting rather than realistic scientific simulation scenarios. There is limited evidence that the method can scale to more complex or real-world physical systems.

**2. Lack of convincing real-world applications**

The evaluation does not include real scientific or engineering applications such as robotics simulation, large-scale CFD systems, or real physical experiments. As a result, the practical impact of the approach remains unclear.

**3. Limited baseline comparisons**

Although the paper includes several baseline models, comparisons with stronger and more recent scientific machine learning methods appear incomplete. It is unclear whether the reported gains hold against more competitive modern approaches.

**4. Performance improvements are modest relative to model complexity**

The architecture introduces multiple components (physics operators, gradient isolation, physics adapter, stacked fusion), yet the improvements over existing methods appear relatively incremental.

**5. Heavy reliance on manually designed physical operators**

The G-KSCH operator basis is manually constructed from known PDE components. The method therefore depends on hand-designed physics priors rather than discovering governing dynamics automatically.

**6. Limited scope of modeled systems**

The current formulation is restricted to dissipative systems defined on regular grids and does not extend naturally to other types of physical systems such as Hamiltonian dynamics, irregular meshes, or action-conditioned environments.

---

> ### Author Rebuttal · Authors · 2026-03-31
>
> We sincerely thank the reviewer for the thoughtful comments and for recognizing the clarity of the dual-stream design, the physically motivated operator construction, and the empirical improvements on selected benchmarks. Below, we clarify several points that may have been unclear in the current presentation.
>
> (1) On scope and realistic scientific modeling
>
> The current paper focuses on state prediction for nonlinear dissipative dynamics on structured grids, which defines its scope. Within this scope, our goal is not to solve large-scale engineering systems end-to-end, but to test whether a reconfigurable physics prior can improve data efficiency and accuracy over learned baselines.
> PDEBench is indeed a simulation benchmark and remains actively used in recent peer-reviewed scientific ML work on PDE modeling (e.g., OmniArch, ICML 2025; WDNO, ICLR 2025). TaxiBJ and WeatherBench are derived from real traffic observations and atmospheric reanalysis data. We will revise the paper to make this distinction explicit and to better state the intended applicability boundary.
>
> (2) On operator sensitivity and reliance on manually designed operators
>
> Our intention is not to claim automatic equation discovery, nor that all operator terms must be rigidly preserved in every domain. APIC uses the physics stream as a structured, reconfigurable prior, while the data stream provides residual flexibility.
>
> The manuscript already includes operator-level ablations in the CFD setting (Table 2), where moving from a physics-agnostic model to NS-based operators and then to the fuller G-KSCH basis yields consistent gains. In addition, following the reviewer’s suggestion, we conducted operator ablations on the traffic task:
>
> Full model: RMSE 17.81, MAE 10.88, MAPE 17.14
>
> No reaction: RMSE 18.10, MAE 11.02, MAPE 17.45
>
> No potential: RMSE 17.96, MAE 10.92, MAPE 17.15
>
> Phase-like only: RMSE 17.92, MAE 10.96, MAPE 17.30
>
> Advection-diffusion only: RMSE 17.92, MAE 10.93, MAPE 17.20
>
> These results suggest that performance degrades gradually rather than catastrophically when terms are removed or modified. APIC does not rely on one brittle hand-crafted formula. Different tasks prefer different subsets or reparameterizations of the operator basis. We will add this analysis to support the interpretation of APIC as a configurable structured prior.
>
> (3) Computational overhead in training and inference
>
> Thank you for raising this point. In the current paper, efficiency is evaluated through VRAM usage, parameter count, and Pareto comparisons.
>
> Figure 1 compares accuracy against VRAM usage and parameter count, while Appendix Figure 6 includes inference-time analysis. At the implementation level, the added physics branch is lightweight: the operator uses fixed finite-difference stencil convolutions with only a small number of learnable coefficients, while gradient isolation is implemented as a stop-gradient/detach operation and does not introduce a heavy forward module. The manuscript also notes that the fourth-order term is implemented by cascading two Laplacians, which reduces memory overhead. On the main CFD benchmark, Table 1 reports that APIC uses 1.36M parameters and 21976 MB VRAM. This is far smaller in parameter count than UNO (6.82M) and TransolverPatched (36.85M), while staying in the same practical resource regime as compact CNN-style backbones. Table 2 also notes that moving from gated fusion to stacked fusion incurs about 0.45M parameters and 5GB VRAM of overhead, while yielding substantial gains.
>
> (4) Generalizability and scope of applicability
>
> The current paper is centered on nonlinear dissipative systems on structured grids, rather than equation discovery in fully unknown domains or broad coverage of all physical regimes. APIC is designed as a reconfigurable neuro-symbolic framework: the physics stream provides a structured operator prior, while the data stream provides residual flexibility, and the operator slot can be adapted across tasks rather than enforcing a fixed equation form. This is already reflected in the manuscript through the task-specific operator adaptations for traffic and meteorology.
>
> The current formulation does not yet cover non-Euclidean domains, irregular meshes, Hamiltonian systems, or action-conditioned dynamics. The present evidence supports applicability within the dissipative structured-grid regime. We additionally evaluated APIC on the explicit PDEBench shock dataset 1D_CFD_Shock_Eta1.e-8_Zeta1.e-8 under a 5% low-data split, comparing Full APIC against FNO, ResNet, and the data-only ablation. The 3-seed validation relative L2 results are: APIC = 0.0112 ± 0.0007, FNO = 0.0150 ± 0.0009, ResNet = 0.0245 ± 0.0070, and Data-only = 0.0119 ± 0.0012. This supports that the framework is not tied to a single 3D benchmark instance and transfers within the same dissipative PDE regime. We will revise the manuscript to make this intended scope and applicability boundary more explicit.

---

> > ### Author Rebuttal · Reviewer_SuXR · 2026-04-03
> >
> > Thank you for the detailed clarifications. The response addresses several of my concerns, and I appreciate the additional analyses and explanations. However, a key issue remains: the lack of qualitative visualization, particularly across diverse and more complex datasets. Visual evidence is important to assess the practical behavior and robustness of the method beyond numerical metrics. If such results are provided, I would be open to increasing my score.

---

> > > ### Author Response · Authors · 2026-04-05
> > >
> > > Thank you for this helpful suggestion. We agree that qualitative evidence is important for assessing practical behavior beyond scalar metrics. Following your suggestion, we have consolidated the qualitative visualizations already included in the manuscript together with the newly added 1D shock results from our previous rebuttal, so that the qualitative evidence can be assessed in a more unified way.
> > >
> > > More specifically, the updated qualitative materials now cover 3D turbulence, WeatherBench, TaxiBJ/traffic, and the newly added 1D shock case, allowing APIC to be visually examined across a broader range of regimes.
> > >
> > > These comparisons include baseline methods and are organized by task. For 3D CFD, we provide multi-view spatial slices and error maps; for traffic, we provide multi-step prediction snapshots; for weather, we provide representative field reconstructions; and for the 1D shock case, we provide both global-profile comparisons and local zoom-ins around the shock region. For convenience, the corresponding qualitative materials are also organized in our anonymous repository: https://anonymous.4open.science/r/APIC-C8C1. We hope this provides a clearer view of the practical behavior and robustness of the method.

---

### Official Review · Reviewer_yZBy · 2026-03-10

**Soundness:** 4
**Presentation:** 2
**Significance:** 3
**Originality:** 4
**Overall Recommendation:** 4
**Confidence:** 3

**Summary:**

The paper presents, Adaptive Physics-Informed Computing (APIC), a neuro-symbolic meta-architecture. The main highlight of the architecture is gradient-isolated interaction strategy. Most of the CNN and Transformer Architectures carry the gradient values or weights from layer to layer where as here isolation helps in avoiding gradient related conflicts. Paper also derived Generalized Kuramato-Sivashinsky-Cahn-Hiliard (G-KSCH) kernel which deals with sparse dynamic Identification. APIC, establishes new benchmarks in 3D Shock wave prediction and performed better than SOTA methods. Additionally, Architecture has been tested on different dataset and thorough ablation study is done before making final conclusions.

**Compliance With Llm Reviewing Policy:**

Affirmed.

**Final Justification:**

It is a good paper and authors have addressed all the concerns raised, I appreciate authors for experimenting SHAP on APIC which can lead to interesting inferences in future. Currently study is restricted to PDE-based spatiotemporal dynamics which I think is limiting the scope of study. As mentioned in the rebuttal to reviewer U67H, please work on wording in the revision so that paper is easy to follow and not filled with overly complex terminologies.

**Key Questions For Authors:**

1) Authors have mentioned in the paper that architecture demonstrate superior robustness in few-shot
scenarios by effectively constraining the hypothesis space via G-KSCH priors with only 10% or 5% data. It would have been interesting to read about zero-shot scenarios. if successful then it would increase the technical depth of the paper.

2) Have authors tried any Explainable AI Technique over APIC Architecture like LIME/SHAP/Grad-CAM because it would be very important to see how gradient-isolated interaction strategy is performing. This might be good addition.

3) Are authors willing to work on more complex systems and try it on ODEs , SDEs and if yes then what kind of modifications would be required. Just small explanation for this would be more than sufficient.

4) Add Impact statement, It was already given in the format and despite of that authors have not included that in the paper.

Explanation to these question would help in better understanding of the paper.

**Limitations:**

Authors have not included Impact Statement. I am not sure why but small impact statement would have been good addition. It is very important for researcher to analyze positive and negative aspect of their work and mention that clearly in the paper. Our main aim is to benefit the community and not just to write paper. This might not be directly related to the society but how one can use this framework and will this require human in loop or to what extent researcher can trust results etc. could have been included.

**Strengths And Weaknesses:**

Here is the thorough assessments of the Strengths and Weakness :

1) Soundness : Paper is technically sound. Claims are supported with proper proofs and experiments. Dedicated section is written on ablation study which is highly important for any research paper. Mathematical explanation is provided and proofs and derivations are mentioned in the Appendix. Very rigorous comparison is done with existing baseline. Very comprehensive paper and appropriate methodology.

2) Presentation : Paper is extremely well written and proper formatting guidelines are followed. Although Impact statement is not included in the paper. Paper involves complex concepts and too many technical terms and hence atleast for me, narrative was not easy to follow. In general for wider audience, with general understanding of ML concepts , Paper is not easy to follow. One must have very crucial knowledge about many topics to understand the paper. Paper also discusses good amount of prior literature but overall makes it hard to follow.

3) Significance : Authors claim to resolve '  tripartite dilemma ' , architecture that addresses accuracy, .efficiency, and physical consistency. Existing Architecture, try to address two of them but the Authors claim and results show that APIC is able to give Pareto Optimal Performance balancing all these criteria. It has also establishes new benchmark for 3D Supersonic Shock waves. In future researcher may try to develop new versions of framework and establish it as benchmark over many complex ODEs and PDEs.

4) Originality : APIC Framework is intelligent idea that decouples physical parameter identification from residual correction. This way it tries to manage spectral bias and inductive bias and provide better results. It is worth to note results are good and with good reasonable margin it is better than SOTA Methods.

---

> ### Author Rebuttal · Authors · 2026-03-31
>
> We sincerely thank the reviewer for the very positive assessment and constructive suggestions. We are glad that the reviewer finds the proposed framework technically sound, the empirical study thorough, and the contribution meaningful.
>
> Regarding the specific questions:
>
> (1) Zero-shot scenarios.
>
> We agree that zero-shot generalization is an interesting and important direction. Our current work focuses on the low-data regime rather than strict zero-shot transfer, and we will clarify this distinction in the final version. We will also avoid over-claiming zero-shot capability and instead position it as a potential extension rather than a demonstrated result. APIC incorporates explicit physics-informed operators as an inductive bias, which may make it more suitable than pure data-driven models for extrapolative settings beyond standard training distributions. We therefore consider zero-shot or near-zero-shot transfer as a natural direction for future work.
>
> (2) Explainability of APIC.
>
> Thank you for this insightful suggestion. Rather than relying on post-hoc explainability tools such as LIME/SHAP, APIC is designed with built-in interpretability through its explicit separation between physics operators and data-driven residuals.
>
> To further examine this aspect, we conducted an additional analysis of the learned adaptive coefficients across multiple random seeds, both with and without gradient isolation (detach). Importantly, we find that explainability in this setting should not be assessed solely based on whether the coefficient variance is numerically smaller, since very small variance may also arise when coefficients collapse toward constrained boundary values or compensation regimes. For example, in Mode-19 and Mode-0595, the no-detach and detached variants can exhibit comparable or even reversed cross-seed standard deviations for some channels/operators, which motivates a more careful interpretation beyond variance alone.
>
> Instead, we focus on whether the learned coefficients preserve the intended role separation between the two streams. In the no-detach setting, some coefficients are more easily influenced by end-to-end residual fitting, so the physics branch may be partially used to absorb errors from the data stream. While this can still yield competitive predictive performance, it makes the learned coefficients less aligned with the intended “physics-trend + data-residual” decomposition. By contrast, with gradient isolation, the physics branch is less directly driven by residual correction, and the learned coefficients tend to better reflect a stable trend-modeling role rather than acting as generic correction terms.
>
> We will clarify this point in the revision and include representative coefficient statistics to make this behavior more transparent.
>
> (3) Extension to ODEs/SDEs.
>
> Thank you for your suggestion. The current framework is designed for PDE-based spatiotemporal dynamics, but the basic idea is not restricted to PDEs in principle. It can be extended to ODE/SDE settings by adapting the operator formulation and discretization scheme. We will briefly discuss this possible extension in the revision.
>
> (4) Impact statement.
>
> We appreciate your suggestion. We acknowledge that the impact statement is missing and will include a proper discussion of both positive impacts and limitations in the final version.
>
> We thank the reviewer again for the encouraging feedback and valuable suggestions.

---

> > ### Author Rebuttal · Reviewer_yZBy · 2026-04-02
> >
> > Thank you for the clarification but I do have few followup Question which I would like authors to answer:
> >
> > 1) Explanation regarding Zero-Shot Scenario sounds fair and make sure to write this down in revision.
> >
> > 2) I agree that LIME/SHAP etc are post-hoc explainability tool but I would like to know if it is possible to apply LIME/SHAP on APIC. It would be interesting to see if post-hoc explainability algorithms give some important inference about APIC. I appreciate the explanation provided by doing additional analysis of the learned adaptive coefficients provides new direction about explainability of APIC. If authors feel it is not possible to use these algorithms like LIME/SHAP on APIC, feel free to write it down with reason.
> >
> > 3) Sounds good regarding the extensions to ODEs/SDEs.
> >
> > 4) It is crucial to read about author's view on positive and negative impact of their work, so can you provide few bullet points regarding same. Bullet points can be later converted into impact statement in the revision.

---

> > > ### Author Response · Authors · 2026-04-05
> > >
> > > Thank you for these helpful follow-up comments. We greatly appreciate these constructive questions and will make sure the corresponding clarifications are reflected in the revision.
> > >
> > > (1) and (3) On the zero-shot setting and possible extensions to ODEs/SDEs
> > >
> > > We will explicitly clarify in the revision that the current paper focuses on low-data settings rather than strict zero-shot transfer, and we will also briefly discuss the possible extension of APIC to ODE/SDE settings.
> > >
> > > (2) On explainability tools
> > >
> > > We also found this suggestion very interesting. Following your question, and within the time constraints of the rebuttal period, we conducted a small SHAP-based experiment to test whether explainability tools can in fact be applied to APIC.
> > >
> > > The results suggest that explainability tools can indeed be applied to APIC, although some care is needed because APIC is a high-dimensional 3D spatiotemporal forecasting model. In our preliminary analysis, we wrapped the model output into a scalar target (based on the pressure channel) and compared the APIC(detach) and APIC(no-detach) variants across multiple random seeds and low-data settings. The results further show that explainability tools are feasible on APIC and reveal different attribution patterns between the two variants, which we find interesting. In particular, across both the 10% and 5% settings, the no-detach model often shows larger global attribution magnitude, while the concentration-related statistics are more mixed. These observations suggest that this type of explainability analysis can provide a useful perspective for understanding differences in how the two variants use input information.
> > >
> > > At the same time, we would like to present this result cautiously. For a model like APIC, the behavior of SHAP depends substantially on the chosen explanation target and aggregation scheme, so we currently view this as a preliminary analysis. More broadly, our current view remains that the interpretability of APIC primarily comes from its built-in structural design, namely the explicit physical operators, the learnable adaptive coefficients, and the analysis of their behavior under detach / no-detach settings; how much additional insight explainability tools can provide for APIC is something we still hope to study further in future work. We sincerely appreciate this suggestion, and we will mention this direction more clearly in the revision and highlight it as an interesting and important avenue for future work.
> > >
> > > (4) On the impact statement
> > >
> > > Following your suggestion, and as the basis for the impact statement that we will add in the revision, we briefly provide the following points:
> > >
> > > 1.Positive impact: For scientific forecasting tasks, APIC may improve data efficiency and physical consistency, and may also reduce training cost to some extent.
> > >
> > > 2.Positive impact: For scientific modeling, the explicit operator basis and adaptive coefficients may make hybrid forecasting models easier to inspect than fully black-box alternatives. This may also allow researchers to replace operator bases and observe prediction/coefficient changes as a form of partial scientific validation or scientific probing.
> > >
> > > 3.Potential limitation / risk: For forecasting tasks, if the chosen operator basis is fundamentally mismatched to the target system, the structured prior may guide the model in an unhelpful direction.
> > >
> > > 4.Potential limitation / risk: Strong benchmark performance should not be over-interpreted as full reliability in real scientific deployment, especially in safety-critical settings.
> > >
> > > 5.Potential limitation / risk: For scientific tasks, human judgment remains important when interpreting predictions, especially under distribution shift or limited observations.
> > >
> > > Finally, we thank the reviewer again for the encouraging and constructive feedback. These suggestions helped us better clarify the scope of the paper, improve the discussion of interpretability, broaden our view of future directions, and better articulate both the potential positive impact and the limitations of the work.

---

### Official Review · Reviewer_U67H · 2026-03-11

**Soundness:** 2
**Presentation:** 1
**Significance:** 3
**Originality:** 1
**Overall Recommendation:** 2
**Confidence:** 4

**Summary:**

The authors propose an architecture in which the evolution of a timestep is prescribed by two parallel architectures. One (which they call a "physical stream") evaluates from a linear combination of standard physics PDEs that can be modeled with finite differences. The other ("Data Stream") concatenates both the state and the output of the physical stream with gradients detached before pushing through a nonlinear transformation. The two "streams" are then combined and transformed a final time. Experiments are provided for a compressible flow with shocks in a simple geometry with comparisons to other methods, and for a traffic flow. There is a very brief discussion regarding weatherbench.

**Compliance With Llm Reviewing Policy:**

Affirmed.

**Key Questions For Authors:**

I don't have any questions, please feel free to correct me if I've misunderstood anything.

**Limitations:**

no suggestions

**Strengths And Weaknesses:**

Soundness: Is the submission technically sound? Are claims well supported (e.g., by theoretical analysis or experimental results)? Are the methods used appropriate? If the paper includes theoretical results, are the proofs correct and based on reasonable assumptions? If the paper includes empirical results, are the experiments well-designed? Are the authors careful and honest about evaluating both the strengths and weaknesses of their work? Note: Soundness is distinct from impact. A paper can be technically sound—meaning correct, rigorous, and methodologically appropriate—even if its contributions are modest or incremental. Conversely, a paper proposing a high-impact idea must still meet the same bar for technical soundness. Reviewers should assess these dimensions separately.

- Originality (poor). The paper does a poor job contextualizing in the broader community. Purely datadriven methods are discussed, and then physics based methods are case as either PINNS (circa pre-2020) or strong constraints. There is a huge range of papers that include the strategy used here. The use of a black-box architecture compensating for the model form error of a model is often referred to as "defect correction" or "differentiable physics". Dictionary based calibration of candidate PDE terms for a PDE is commonplace, starting with SINDy-like approaches but including hybrid techniques that do both defect correction and dictionary learning like the current work.

- Presentation (poor). The overuse of buzzwords and pseudo-technical phrases that lack a specific meaning make this paper unneccesarily hard to read. Everything has a name - why call something "stacked fusion" rather than concatenating outputs? Gradient isolation strategy = we turned off gradient tracking. "Furthermore, capitalizing on the Eulerian structural isomorphism between fluid dynamics and macroscopic traffic flow..." is another meaningless statement (they are both conservation laws, that is not an isomorphism in any mathematiclaly rigorous sense of the word). The paper would be significantly strengthened without this.

- Soundness (poor). The main claim, that gradients are orthogonal, is not true. That would imply $\nabla_theta L_phys \cdot \nabla_theta L_data = 0$. This is not true - by turning off gradients they just block the gradient update from being accounted for. This is standard in many RL methods and a useful trick (and may be important for their performance), but along with many parts of the rest of the paper dressing up routine techniques like a detached gradient with buzzwords makes it very difficult to read and to take seriously.

- Significance (potentially strong). The empirical studies suggest that there is a meaningful improvement on the state of the art. An ablation study clearly articulates that the various ingredients are more than the sum of their parts. Multiple metrics on varying field values, rather than lumping everything into MSE, is important. There is substantial parameter efficiency demonstrated compared to the state of the art. While the idea is not particularly novel in my opinion, they seem to have come up with good choices of architecture and gotten something performant to come out.

- Weak benchmarks. The problems considered have relatively simple physics that are well captured by their dictionary. If they wanted to truly push the method they should consider problems that cannot be naturally modelled by simple PDE terms - e.g. fractional/nonlocal transport, stochastic physics, Riemann problems with very strong shocks (focusing on transonic dodges the issue that stencils will not be informative with discontinuities).

---

> ### Author Rebuttal · Authors · 2026-03-31
>
> We thank the reviewer for the rigorous critique. We appreciate the recognition that our empirical study shows meaningful improvement, uses multiple evaluation metrics, and demonstrates strong parameter efficiency. We agree that the original manuscript overstates parts of the method in terminology, framing, and positioning, and we will revise it substantially. Here we clarify four central points:
>
> (1) On terminology and positioning.
>
> In the revision, we will replace claims with more precise wording: “gradient-orthogonalized mechanism” and “mechanistically decoupled optimization paths” will be revised to “gradient isolation” and “reduced interference between optimization paths”; “stacked fusion” will be replaced by “channel concatenation followed by nonlinear transformation”; “alignment-stacking-decoupling” will be replaced by “feature alignment, concatenation, and stop-gradient-based isolation”; “universal operator” will be softened to “reconfigurable operator basis”; “cross-domain generalization” to “cross-task adaptability”; and “few-shot” to “low-data regime.” We will also remove unnecessary theoretical embellishments and remaining buzzword-like phrasing.
>
> (2) On stop-gradient.
>
> We agree that stop-gradient does not imply strictly orthogonal gradients, and our original wording was strong. Detach removes the dominant backward path from the data-fitting branch to the adaptive coefficients of the explicit-prior branch. The intended claim is progressive/pathwise decoupling that weakens direct cross-branch interference during end-to-end training. The use of detach is not the source of APIC’s accuracy, as shown by experimental results where APIC(nodetach) and APIC(detach) have very similar L2. Without detach, the coefficients of the operator basis are easily pushed toward the boundaries of their feasible range and become unstable across different random seeds. With gradient isolation, the gradient influence from the data-driven branch on the coefficients of the physical branch is avoided, making the coefficients significantly more stable across seeds. So we note that our original wording used the process-oriented term “orthogonalized” rather than a claim of strict mathematical orthogonality but we agree that the surrounding presentation made this sound stronger than intended, and we will revise it accordingly.
>
> (3) On relation to defect-correction and dictionary-learning.
>
> We thank the reviewer for pointing out the broader context of defect-correction, differentiable-physics, and dictionary-based hybrids (e.g., SINDy-style approaches). We agree that APIC should be positioned relative to this literature, and we will revise Related Work accordingly. In classical defect-correction, a physics solver is typically the primary simulator and a learned module patches its output. In dictionary-learning / equation-discovery, the goal is often to recover an explicit governing equation. By contrast, APIC is an end-to-end forecasting architecture in which the data-driven backbone remains the primary predictive component, while the explicit G-KSCH branch functions as an in-network structured prior that constrains the hypothesis space during prediction rather than recovering the final equation itself. APIC also differs from simple output-level residual correction: instead of adding a learned patch to a completed physical prediction, it projects the explicit physical state into latent space and fuses it with learned features through concatenation and nonlinear transformation. We therefore characterize APIC as a hybrid forecasting architecture with a reconfigurable structured prior.
>
> (4) On benchmark difficulty.
>
> We thank the reviewer for raising benchmark difficulty. We first correct an error in the manuscript: the main PDEBench case was mistakenly written as $M=0.8$, whereas the actual dataset used is \texttt{3D\_CFD\_Turb\_M1.0\_Eta1e-08\_Zeta1e-08\}. This matters because the core benchmark is already a nontrivial 3D compressible turbulence/shock prediction task. To further address the reviewer’s concern, and within the remaining rebuttal time budget, we additionally evaluated APIC on the explicit PDEBench shock dataset \texttt{1D\_CFD\_Shock\_Eta1.e-8\_Zeta1.e-8\} under a 5\% low-data split, comparing Full APIC against FNO, ResNet, and the data-only ablation. The 3-seed results are: APIC = $0.0112 \pm 0.0007$, FNO = $0.0150 \pm 0.0009$, ResNet = $0.0245 \pm 0.0070$, and Data-only = $0.0119 \pm 0.0012$ in validation relative L2. These additional results directly test a more explicit shock setting and show that APIC achieves the best mean performance, outperforming. Finily We agree that stochastic, fractional/nonlocal, or more extreme shock regimes would be valuable future tests.

---

> > ### Author Rebuttal · Reviewer_U67H · 2026-04-03
> >
> > Thank you for making the effort to improve the submission. While these changes are likely to improve the quality of the manuscript, the quantitative implications of the final result are of less impact. I would raise my score to a marginal reject.

---

> > > ### Author Response · Authors · 2026-04-05
> > >
> > > Thank you for the follow-up and for acknowledging that the revision would improve the manuscript. We appreciate your updated assessment.
> > >
> > > We understand that the remaining concern is primarily about how the method should be interpreted. In the revised manuscript, we will make clearer that APIC serves as a structured prior within an end-to-end forecasting architecture, and we will sharpen the framing accordingly.
> > >
> > > In addition, to make the empirical behavior of the method more transparent, we have also consolidated the qualitative evidence and broadened its presentation across multiple regimes. The updated qualitative results now cover 3D turbulence, WeatherBench, TaxiBJ/traffic, and the added 1D shock case in our anonymous repository: https://anonymous.4open.science/r/APIC-C8C1., including multi-view CFD slices and error maps, multi-step traffic predictions, representative weather reconstructions, and global/local shock-profile comparisons. We hope this helps make the practical behavior of the method easier to assess beyond the scalar metrics alone.
> > >
> > > We fully respect your concerns regarding originality and framing, and we will ensure that the manuscript is revised thoroughly to address them. We appreciate your consideration and hope the revised version will be viewed more favorably in light of these revisions.

---

### Official Review · Reviewer_M76G · 2026-03-13

**Soundness:** 2
**Presentation:** 3
**Significance:** 2
**Originality:** 3
**Overall Recommendation:** 4
**Confidence:** 3

**Summary:**

Current physics models either are biased in frequency space (purely data-driven) or suffer from optimization difficulties (physics constrained).
APIC uses gradient isolation to decouple optimization of parameter identification and residual correction reducing gradient conflicts.
The method implements known governing equations for nonlinear dissipative system as prior operators

The paper develops an architecture with a physics and a datastream. The physics stream uses a partial differential equation over space and time to represent conservation over a discretized field with discretized derivatives with a parameterized operator representing nonlinear advection, fickian diffusion, phase separate,hyper diffusion and reaction source which are various dissipative structures appearing in non-equlibrium thermodynamics.

The architecture processes the data and physics stream in parallel with isolated gradients. The physics stream derives a coarse estimate. direct interaction suffers from alignment problems which is solved by gradient isolation by blocking gradients and manifold alignment where both streams are projected onto a shared space. The physical stream and the data stream follow orthogonal optimization paths eliminating gradient conflicts in hybrid modeling. This improves over gating mechanisms for combining streams.

**Compliance With Llm Reviewing Policy:**

Affirmed.

**Final Justification:**

My main concerns were over establishing the usefulness of gradient isolation which i think has been addressed by the rebuttal. A full discussion of this would be useful in the paper. Reading other reviews I also think that stronger benchmarks and baselines would make the demonstration more convincing. I think my current score is accurate.

**Key Questions For Authors:**

How does mixing disparate thermodynamical operators lead to a sensible operator for modeling physics? What is the justification for mixing these terms?

Are the physics and data stream separately trainable? Is there any advantage to training the two streams jointly giving the strong gradient isolation? Couldn’t the physics stream be trained separately and then it would serve as an initialization for the data stream?

Line 263: do you have references for models where gating is employed leading to gradient conflicts and/or poor performance? Do you have evidence or ablations demonstrating spatiotemporal semantic misalignment (Line 95)?

Line 293 (col 2). What do you mean by the few-shot regime? Are you doing few-shot learning or Is this the same as low-data regime?

In what sense is the experiment in 4.3 performing cross-domain generalization?

**Limitations:**

Yes.

**Strengths And Weaknesses:**

The paper targets hybrid modeling where physical prior knowledge information in the form of a flexible PDE operator is combined with data-driven processing. The operator covers various forms of dissipative dynamics built on knowledge of thermodynamics.

The method is evaluated on 3  datasets including CFD and weather/climate data. A number of dynamical baselines are used for comparison. The method appears to perform better than well-known baselines especially in the low-data regime with up to initial 5% of the sequence, showing data efficiency.

Ablation experiments evaluate the architectural features of stacking v. gating.

The introduction section in the paper is not very clear (75-107). However, the description in the methodology improves the presentation.

The strong isolation of the streams, I think, is not fully justified.

 Second it is not clear whether in the presence of such isolation, whether it is essential to do joint training? Further I do not see an experiment that performs an ablation on the physics and data streams in isolation. How much of an advantage do the streams confer individually and in combination is not entirely clear.

Furthermore, the model’s performance under noisy settings is not explored. A limitation of finite difference gradient calculations is that are not very robust to noise and often require high-fidelity measurements


Typo: line 58 col 2. conflicts neural …

---

> ### Author Rebuttal · Authors · 2026-03-31
>
> We sincerely thank the reviewer for the thoughtful and constructive comments. We especially appreciate the reviewer’s questions on stream isolation, joint training, and robustness under noise, as these directly point to the most important clarifications needed for the paper. Following these suggestions, we conducted additional ablations and noise-robustness analyses.
>
> (1) Why does mixing multiple thermodynamical operators lead to a sensible operator?
>
> Our object is not to claim that all listed dissipative terms must literally and simultaneously govern every target system, nor to present APIC as a fixed PDE solver. Instead, these terms define a structured operator basis for dissipative dynamics: the physics branch provides a constrained trend space shaped by human-derived operators, while adaptive coefficients and the data branch specialize within that space. We will revise the presentation to avoid over-claiming universality.
>
> (2) Are the physics and data streams separately trainable? Is joint training still necessary?
>
> The two streams are separately trainable, but not separately sufficient. We performed the requested ablation under a unified 3-seed protocol.
> 10% split: DataOnly = 0.0776 ± 0.0029, PurePhysics = 0.3193 ± 0.0000, APIC(no-detach) = 0.0498 ± 0.0012, APIC(detach) = 0.0502 ± 0.0002.
> 5% split: DataOnly = 0.0926 ± 0.0056, PurePhysics = 0.3336 ± 0.0000, APIC(no-detach) = 0.0599 ± 0.0009, APIC(detach) = 0.0616 ± 0.0015.
> These results demonstrated a clear pattern: physics-only underfits badly, data-only is consistently weaker than APIC, and the dual-stream design remains essential in low-data settings. Although stop-gradient blocks the dominant backward path from the data loss to the physics coefficients, the two branches still co-adapt through the shared prediction target and fusion pathway; this is different from a stage-wise pipeline where the physics branch is trained once and then frozen.
>
> (3) What is the role of gradient isolation?
>
> We agree that “orthogonal optimization paths” was overly strong. More precisely, stop-gradient removes the dominant indirect Jacobian path from the data branch to the physics coefficients, leaving only shallow fusion interaction. We will revise the wording accordingly.
>
> In particular, the main benefit is not a large accuracy gain: no-detach and detach have very similar L2 performance above. Rather, the difference lies in how the adaptive coefficients are used. Without detach, coefficients are more easily driven by residual fitting, so the physics branch can be partially co-opted into a correction pathway. With detach, this is suppressed, and the intended “physics-trend + data-residual” role separation is better preserved. We will clarify this distinction in the revision.
>
> (4) Noise robustness.
>
> We thank the reviewer for highlighting this limitation. We conducted eval-time Gaussian noise experiments at 1%, 5%, and 10%. All models degrade, consistent with the sensitivity of finite-difference schemes, but the degradation is gradual and APIC consistently outperforms DataOnly across both L2 and H1.
>
> 10% split: DataOnly L2 changes from 0.0834 → 0.0834 → 0.0845 → 0.0877 as noise increases from 0% to 10%, while APIC(detach) changes from 0.0505 → 0.0505 → 0.0518 → 0.0560.
>
> 5% split: DataOnly changes from 0.1131 → 0.1132 → 0.1137 → 0.1154, while APIC(detach) changes from 0.0619 → 0.0619 → 0.0630 → 0.0679.
>
> The same trend holds for H1. Thus, while APIC is not immune to observation noise, it degrades gracefully and maintains a clear advantage over the data-only baseline.
>
> (5) Gating / semantic misalignment.
>
> We agree the original presentation should be better supported. We will add references and improve the presentation to avoid overstatement.
>
> (6) Few-shot and cross-domain terminology.
>
> We used “few-shot” informally to refer to low-data regimes (5%/10%), not meta-learning; we will correct this. We will also revise “cross-domain generalization” to a more accurate transferability/adaptability wording if needed.
> We will also improve the introduction for clarity and fix the noted typo. We thank the reviewer again for the helpful suggestions, which directly improved both our analysis and presentation.

---

> > ### Author Rebuttal · Reviewer_M76G · 2026-04-02
> >
> > Thank you for the response. The joint v. separate training experiment setup I think is not entirely convincing. The two streams could still be trained separately and then a small adapter could conceivably still be trained to adapt the networks. I think the strong separation requires a detailed experiment and analysis to establish that this is essential.
> >
> > I also agree with the other reviewers (SuXR, U67H) that the experimental tasks lack in complexity and real-world relevance.

---

> > > ### Author Response · Authors · 2026-04-05
> > >
> > > Thank you very much for this precise and constructive follow-up, especially the suggestion to test a stage-wise alternative in which the two branches are trained separately and then connected by a lightweight adapter. We appreciate this suggestion and therefore conducted exactly this additional ablation.
> > >
> > > Specifically, under the low-data 10% and 5% settings, we first trained the data system and the physics system separately, then froze both systems and trained only a small 3-layer 3D convolutional adapter on top of their concatenated outputs. We chose this design to be lightweight but still sufficiently expressive, so that it constitutes a fair “small adapter” test rather than a heavily redesigned fusion model.
> > >
> > > The results support the reviewer’s hypothesis as a meaningful alternative to test, but they also show that this stage-wise strategy is not sufficient to recover the performance of jointly trained APIC. The validation relative L2 results of the stage-wise + adapter variant are:
> > >
> > > 5% split: 0.1154 / 0.1144 / 0.1125 across seeds 43 / 2019 / 2026, i.e. 0.1141 ± 0.0012
> > >
> > > 10% split: 0.1026 / 0.1036 / 0.1034 across seeds 43 / 2019 / 2026, i.e. 0.1032 ± 0.0004
> > >
> > > The trends for H1, divergence, and vorticity are consistent with the L2 results. In particular, this stage-wise variant remains clearly below APIC(detach), and in practice is not competitive with the full jointly trained model. This suggests that the benefit of APIC does not come merely from the coexistence of a physics system and a data system plus a lightweight post-hoc fusion module. Rather, even under gradient isolation, end-to-end co-adaptation under a shared predictive objective remains important for learning a useful division of labor between the physics system and the data system.
> > >
> > > Regarding the reviewer’s concern on task complexity and real-world relevance, we fully agree that our current study is benchmark-based rather than a direct deployment study on sensor-driven real-world CFD pipelines. At the same time, we would like to clarify the intended scope more carefully in the revision. PDEBench is indeed a simulation benchmark, but it remains actively used in recent peer-reviewed scientific ML work on PDE modeling (e.g., OmniArch, ICML 2025; WDNO, ICLR 2025), partly because dense spatiotemporal ground truth is rarely available in real physical systems. In addition, TaxiBJ and WeatherBench are derived from real traffic observations and atmospheric reanalysis data, respectively. To further address the reviewer’s concern about complexity beyond the main 3D CFD setting, we also evaluated APIC on the explicit PDEBench shock dataset 1D_CFD_Shock_Eta1.e-8_Zeta1.e-8 under a 5% low-data split. The 3-seed validation relative L2 results are: APIC = 0.0112 ± 0.0007, FNO = 0.0150 ± 0.0009, ResNet = 0.0245 ± 0.0070, Data-only = 0.0119 ± 0.0012.
> > >
> > > We are sincerely grateful for this highly constructive suggestion. It directly helped us identify a more precise alternative hypothesis, and we believe the new stage-wise + adapter ablation substantially strengthens the evidence for why joint training remains important in APIC.

---

### Decision · Program_Chairs · 2026-04-30

**Decision:**

Accept (regular)

**Comment:**

This paper received mixed opinions from all reviewers. The main technical issues raised by reviewers include the justification of both the design and the efficacy of the method's key components, demonstrations with stronger baselines and benchmark tests with more complex physics, and the paper's novelty relative to prior work.

The rebuttal discussion helped to address some of these concerns and ended with two positive and two negative reviewers. One negative reviewer (Reject) did not provide their final justification and expressed that they would raise their score to a marginal reject (see Rebuttal Acknowledgment). The other negative reviewer indicated that the additional results in the rebuttal solved their concern (see Final Justification). Therefore, I discounted the negative opinions and recommend that ICML accept this work. Still, I feel this is a borderline case, and I would not mind if SAC/PC overturns my recommendation.